# Amortising Inference and Meta-Learning Priors in Neural Networks

**Tommy Rochussen**[1,2,3] **& Vincent Fortuin**[1,2,4]
[1] Helmholtz AI, [2] Munich Center for Machine Learning, [3] Technical University of Munich,
[4] University of Technology Nuremberg.
{thomas.rochussen, vincent.fortuin}@helmholtz-munich.de

## Abstract

One of the core facets of Bayesianism is in the updating of prior beliefs in light of new evidence—so how can we maintain a Bayesian approach if we have no prior beliefs in the first place? This is one of the central challenges in the field of Bayesian deep learning, where it is not clear how to represent beliefs about a prediction task by prior distributions over model parameters. Bridging the fields of Bayesian deep learning and probabilistic meta-learning, we introduce a way to *learn* a weights prior from a collection of datasets by introducing a way to perform per-dataset amortised variational inference. The model we develop can be viewed as a neural process whose latent variable is the set of weights of a BNN and whose decoder is the neural network parameterised by a sample of the latent variable itself. This unique model allows us to study the behaviour of Bayesian neural networks under well-specified priors, use Bayesian neural networks as flexible generative models, and perform desirable but previously elusive feats in neural processes such as within-task minibatching or meta-learning under extreme data-starvation.

## 1 Introduction

While the ability to learn hierarchical representations of data has allowed neural networks to boast phenomenal predictive performance across many domains, enabling them to accurately estimate their predictive uncertainty remains generally unsolved. Bayesian deep learning (MacKay, 1992) promises a theoretically sound approach for endowing representation learners with uncertainty quantification, but there are many difficulties with the approach in practice. Of all of them, one of the most sleep-depriving is the question of how to choose appropriate priors. Neural network weights lack interpretability, meaning it is fiendishly difficult to elicit priors over them that are sensible in prediction space (Fortuin, 2022). The convenience priors that we generally use are known to reduce large[1] Bayesian neural networks (BNNs) to the "simple smoothing devices" (MacKay, 1998) that are Gaussian processes (GPs; Neal, 1995; Matthews et al., 2018; Yang, 2019), meaning that in trying to achieve uncertainty quantification we inadvertently destroy the ability to learn hierarchical representations (Aitchison, 2020). An increasingly popular approach to specifying priors in BNNs is to use function-space priors (Flam-Shepherd et al., 2017; Sun et al., 2019; Cinquin et al., 2024). However, these priors are most often chosen to be GP priors which, yet again, reduce Bayesian deep learning to approximate GP inference. Even when priors and architectures are chosen specifically to avoid GP behaviour, the resulting stochastic process prior is not just poorly understood, but generally a bad model of the real-world data-generating process (Karaletsos and Bui, 2020).

On the other hand, neural processes (NPs; Garnelo et al., 2018a;b) use the shared structure between related tasks to meta-learn a free-form stochastic process prior. Unlike GPs and BNNs, they do not learn an explicit parametric prior that can be evaluated or sampled from, but, given some context observations, they learn to map directly to the stochastic process posterior corresponding to the learned implicit prior process. For a sufficiently flexible NP architecture and with enough data, they can model the ground-truth data-generating process (Foong et al., 2020). Given this remarkable ability, neural processes have enabled practitioners to *endow representation learners with uncertainty quantification* to great success across a range of domains including weather and climate applications (Allen et al., 2025; Ashman et al., 2024b; Andersson et al., 2023), causal machine learning (Dhir

---

[1]In the sense of infinite architecture width, architecture depth, number of channels, or so on.

et al., 2025), Bayesian optimisation (Maraval et al., 2023; Volpp et al., 2023), and cosmological applications (Park and Choi, 2021; Pondaven et al., 2022).

Drawing inspiration from neural processes, we are interested in meta-learning BNN priors that encode the shared structure across a related set of tasks. In other words, *can we use meta-learning to design well-specified priors in Bayesian deep learning?* To that end, **we devise a scheme for performing per-dataset amortised inference in BNNs**. This results in a neural process whose latent variable is the set of weights of a BNN, and whose decoder is the neural network parameterised by a particular latent variable posterior sample. We refer to our model as the Bayesian neural network process (BNNP). The unique ability of the BNNP to amortise BNN inference and meta-learn BNN priors **enables us to investigate such previously unanswerable questions** as: "*under a well-specified prior, how important is the approximate inference method in Bayesian deep learning really?*". Furthermore, as a new member of the neural process family (Dubois et al., 2020), the BNNP introduces some completely novel capabilities. These include the ability to perform ***within-task* minibatching** for scalability to massive context sets, and the ability to **adjust the flexibility of the learned prior** so that overfitting can be avoided in settings for which only a few tasks have been observed.

## 2 THE BAYESIAN NEURAL NETWORK PROCESS

### 2.1 LAYERWISE INFERENCE

We start by considering the conditional posterior over the last-layer weights $\mathbf{W}^L \in \mathbb{R}^{d_{L-1} \times d_L}$ in an $L$-layered multilayer perceptron (MLP), where $d_l$ denotes the number of units in the $l$-th layer of the network. In this exposition we do not consider biases but they are straightforward to include in practice. Given some data $\mathcal{D} = \{\mathbf{X}, \mathbf{Y}\}$ where $\mathbf{X} \in \mathbb{R}^{n \times d_0}$ and $\mathbf{Y} \in \mathbb{R}^{n \times d_L}$ as well as the weights of the previous layers $\mathbf{W}^{1:L-1} = \{\mathbf{W}^l\}_{l=1}^{L-1}$, and assuming a Gaussian likelihood, the conditional posterior over the last-layer's weights is of the form

$$p(\mathbf{W}^L | \mathbf{W}^{1:L-1}, \mathcal{D}) \propto p(\mathbf{W}^L) \prod_{d=1}^{d_L} \prod_{n=1}^{N} \mathcal{N}\Big(y_{n,d}; \phi(\mathbf{x}_n^{L-1})^\top \mathbf{w}_d^L, \sigma_d^2\Big) \tag{1}$$

where $y_{n,d}$ is the $d$-th dimension of the $n$-th target, $\phi(\cdot)$ is the MLP's elementwise nonlinearity, $\mathbf{x}_n^{L-1}$ denotes the activations of the penultimate layer for the $n$-th datapoint, $\mathbf{w}_d^L$ is the $d$-th column of $\mathbf{W}^L$ representing all input weights to unit $d$ in the output layer, and $\sigma_d$ is the observation noise level for the $d$-th dimension of the targets. Assuming that the prior $p(\mathbf{W}^L)$ is conjugate, this posterior is available in closed-form via Bayesian linear regression (BLR; Bishop, 2007). Inspired by this result, Ober and Aitchison (2021) propose a variational posterior for BNNs and deep GPs that decomposes into a product of layerwise conditionals

$$q(\mathbf{W}) = \prod_{l=1}^{L} q(\mathbf{W}^l | \mathbf{W}^{1:l-1}) \tag{2}$$

where the layerwise factors are computed via exact inference between the layerwise prior and a set of variationally-parameterised pseudo-likelihood terms. We adopt a similar approach.

### 2.2 AMORTISING LAYERWISE INFERENCE

We generalise the likelihood of the last-layer weights seen in Eq. (1) to the weights of the $l$-th layer as follows

$$p(\mathbf{Y}^l | \mathbf{X}^{l-1}, \mathbf{W}^l) = \prod_{d=1}^{d_l} \prod_{n=1}^{N} \underbrace{\mathcal{N}\Big(y_{n,d}^l; \phi(\mathbf{x}_n^{l-1})^\top \mathbf{w}_d^l, {\sigma_{n,d}^l}^2\Big)}_{t_n(\mathbf{w}_d^l)} \tag{3}$$

where $y_{n,d}^l$ is a pseudo-observation corresponding to the $d$-th activation of the $l$-th layer for the $n$-th datapoint, and $\sigma_{n,d}^l$ is the noise level for the corresponding pseudo-likelihood term. Note that $\mathbf{X}^0 \equiv \mathbf{X}$. These two parameters are obtained by passing the $n$-th input-output pair through an inference network $g_{\theta_l}$:

$$\mathbf{y}_n^l, \log(\boldsymbol{\sigma}_n^l) = g_{\theta_l}(\mathbf{x}_n, \mathbf{y}_n) \tag{4}$$

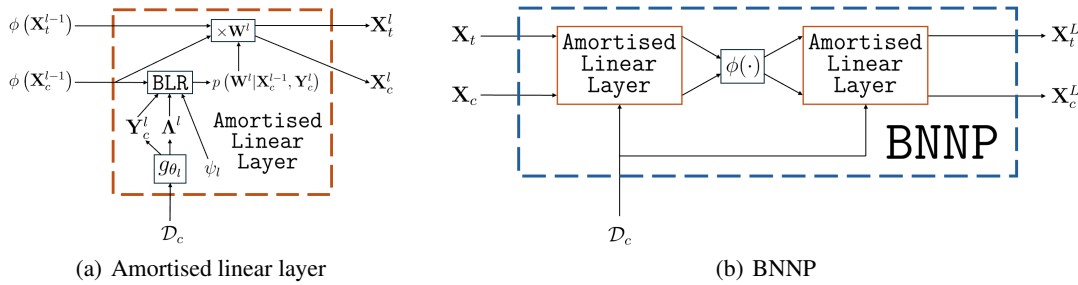

(a) Amortised linear layer                    (b) BNNP

Figure 1: Computational diagrams of (a) the amortised linear layer, and (b) a BNNP with one hidden layer of activations. We use the context $\cdot_c$ and target $\cdot_t$ notation to distinguish between inputs with labels, on which we condition, and inputs without labels, at which we predict.

where $\theta_l$ denotes the parameters of the $l$-th layer's inference network, which will be optimised during training[2]. In the case of the final layer we can use the actual observations, that is, $\mathbf{Y}^L \equiv \mathbf{Y}$. With the pseudo-likelihood parameters, and assuming unitwise-factorised Gaussian priors $p_{\psi_l}(\mathbf{W}^l) = \prod_{d=1}^{d_l} \mathcal{N}\left(\mathbf{w}_d^l; \boldsymbol{\mu}_d^l, \boldsymbol{\Sigma}_d^l\right)$ where $\psi_l = \{\boldsymbol{\mu}_d^l, \boldsymbol{\Sigma}_d^l\}_{d=1}^{d_l}$, the approximate layerwise posteriors are computed in closed form:

$$q(\mathbf{W}^l|\mathbf{W}^{1:l-1}, \mathcal{D}) = p(\mathbf{W}^l|\mathbf{X}^{l-1}, \mathbf{Y}^l) \propto \prod_{d=1}^{d_l} \mathcal{N}\left(\mathbf{w}_d^l; \boldsymbol{\mu}_d^l, \boldsymbol{\Sigma}_d^l\right) \mathcal{N}\left(\mathbf{y}_d^l; \phi(\mathbf{X}^{l-1})^\top \mathbf{w}_d^l, \boldsymbol{\Lambda}_d^{l^{-1}}\right)$$

$$= \prod_{d=1}^{d_l} \mathcal{N}\left(\mathbf{w}_d^l; \mathbf{m}_d^l, \mathbf{S}_d^l\right) \tag{5}$$

where $\boldsymbol{\Lambda}_d^l \in \mathbb{R}^{N \times N}$ is a diagonal precision matrix with the $n$-th diagonal element given by $\frac{1}{\left(\sigma_{n,d}^l\right)^2}$, and the posterior mean vectors $\mathbf{m}_d^l$ and covariance matrices $\mathbf{S}_d^l$ are given by

$$\mathbf{S}_d^{l^{-1}} = \boldsymbol{\Sigma}_d^{l^{-1}} + \phi(\mathbf{X}^{l-1})^\top \boldsymbol{\Lambda}_d^l \phi(\mathbf{X}^{l-1}) \tag{6}$$

$$\mathbf{m}_d^l = \mathbf{S}_d^l \left(\boldsymbol{\Sigma}_d^{l^{-1}} \boldsymbol{\mu}_d^l + \phi(\mathbf{X}^{l-1})^\top \boldsymbol{\Lambda}_d^l \mathbf{y}_d^l\right). \tag{7}$$

We consider inference under Gaussian priors with different factorisation structures in Appendix B.

This machinery, which we call the *amortised linear layer*, enables inference over the weights of a linear layer that is situated arbitrarily within a neural network architecture, conditional on the weights of the previous layers. By stacking these layers and sampling from each layer's conditional posterior before computing the next layer's posterior, we can perform amortised inference in a BNN via the variational posterior

$$q\left(\mathbf{W}|\mathcal{D}\right) = \prod_{l=1}^{L} q\left(\mathbf{W}^l|\mathbf{W}^{1:l-1}, \mathcal{D}\right) \tag{8}$$

$$= \prod_{l=1}^{L} p\left(\mathbf{W}^l|\mathbf{X}^{l-1}, \mathbf{Y}^l\right). \tag{9}$$

Since such a model may be interpreted as a latent-variable neural process (Garnelo et al., 2018b) in which the latent variable is the set of BNN weights and the decoder is the BNN itself, we refer to this model as the *Bayesian neural network process* (BNNP). See Fig. 1 for computational diagrams of the amortised linear layer and a BNNP.

## 2.3 TRAINING

We adopt a variational approach to training, where the parameters to be optimised are the inference network parameters $\Theta = \{\theta_l\}_{l=1}^{L}$ and the prior parameters $\Psi = \{\psi_l\}_{l=1}^{L}$. The former take the role

---

[2]Another option is to pass the concatenation of an input-output pair *as well as* the previous layer's activation $\mathbf{X}_n^{l-1}$ into each inference network. It is possible this could allow for smaller inference networks.

of variational parameters while the latter are model parameters. We motivate our choice of training objective by considering what behaviour we would like to encourage in the BNNP. We summarise these considerations into the following three desiderata:

(I) accurate approximate posteriors,

(II) a prior that faithfully encodes the data-generating process,

(III) high quality predictions.

Assuming access to a meta-dataset $\Xi = \{\mathcal{D}^{(j)}\}_{j=1}^{|\Xi|}$ of tasks that share a data-generating process $\mathcal{D}^{(j)} \sim p(\mathcal{D})$, and partitioning datasets into disjoint context $\cdot_c$ and target $\cdot_t$ sets, we propose a novel objective function for training the BNNP. We call it the posterior-predictive amortised variational inference (PP-AVI) objective, and it is defined as

$$\mathcal{L}_{\text{PP-AVI}}(\Xi) := \frac{1}{|\Xi|} \sum_{j=1}^{|\Xi|} \log q\left(\mathbf{Y}_t^{(j)} | \mathcal{D}_c^{(j)}, \mathbf{X}_t^{(j)}\right) + \mathcal{L}_{\text{ELBO}}\left(\mathcal{D}_c^{(j)}\right) \tag{10}$$

where the first term is the log posterior-predictive density of the target set, and $\mathcal{L}_{\text{ELBO}}(\mathcal{D})$ denotes the usual evidence lower bound (ELBO) used for VI in BNNs. The first term in isolation is the usual objective for NPs, reflecting that NP users often only care about predictions (Foong et al., 2020). Since we intend to use the BNNP to study BNNs, we introduce the ELBO term to explicitly encourage high-quality approximate inference.

**Proposition 1.** *For $|\Xi| \to \infty$, maximisation of $\mathcal{L}_{PP\text{-}AVI}(\Xi)$ directly targets the three desiderata.*

A proof of Proposition 1 involving formal definitions of our desiderata, a practical guide to the PP-AVI objective, and a discussion relating it to other objectives in the literature are provided in Appendix A. We emphasise here, though, that $\mathcal{L}_{\text{PP-AVI}}(\Xi)$ can be unbiasedly estimated from a minibatch of tasks $\xi \subset \Xi$ via $\mathcal{L}_{\text{PP-AVI}}(\xi)$, enabling training on huge meta-datasets via stochastic optimisation.

## 2.4 WITHIN-TASK MINIBATCHING VIA SEQUENTIAL BAYESIAN INFERENCE

When making predictions on a particular task, the full set of real and pseudo observations must be stored in memory in order to compute the posterior. So, while stochastic optimisation allows us to scale to large *meta*-datasets, scalability to large *context* sets remains elusive. Fortunately however, we provide a solution to this problem too. The high memory requirements can be avoided by iteratively updating each layer's posterior with a minibatch of datapoints via sequential Bayesian inference (Bishop, 2007), temporarily discarding each minibatch after applying its update to a particular layer's conditional posterior.

We partition a task $\mathcal{D}$ into $B$ minibatches $\{\mathcal{D}_b\}_{b=1}^B$. We use $\mathbf{W}_{q_b}^l$ to denote a sample from the $l$-th layer's posterior given only minibatch $b$, that is, $\mathbf{W}_{q_b}^l \sim q(\mathbf{W}^l | \mathbf{W}^{1:l-1}, \mathcal{D}_b)$. Since computation of each layer's posterior requires samples from the *full-batch* previous layer posteriors (i.e., $\mathbf{W}_{q_{1:B}}^{1:l-1}$), the minibatching procedure must be performed in full for each layer before sampling and proceeding to the next layer. The ability to minibatch a forward pass over a given context set is rare in the context of neural processes, and it is a valuable property as it allows us to scale to tasks with large and high-dimensional datasets without running into memory limitations. Although some extra stochasticity is incurred when training, at prediction time our minibatching procedure results in the exact same approximate posterior as the full-batch procedure. See Appendix C for a detailed algorithmic description of the procedure and further discussion including complexity analysis of the BNNP. In Appendix D we use a similar sequential inference trick to devise an online-learning scheme for the BNNP through which predictions for a given task can be updated in light of new data.

## 2.5 ADJUSTING THE FLEXIBILITY OF THE PRIOR

One limitation shared amongst neural processes is their poor performance when the number of observed tasks is limited (Rochussen and Fortuin, 2025). This happens because the model parameters are responsible for both amortising (predictive) inference *and* encoding a prior, such that there is no way to discourage the learned prior from overfitting without also affecting the model's ability to

amortise prediction. In the BNNP however, these roles are disentangled into two distinct parameter groups; those of the weights prior, and those of the inference networks. This separation enables us to limit the flexibility of the learned prior *without* affecting inference. We do this by fixing the prior over a subset of the weights and optimising only the remaining prior parameters. For example, we can fix the prior over the last layer's weights to a zero-centered and unit-variance diagonal Gaussian while optimising the prior parameters for the earlier-layer weights. By varying the number of weights whose prior is fixed, we introduce a new knob with which to tune the flexibility of the learnable prior. Such a scheme enables practitioners to balance between forcing a broad/misspecified prior and learning an overfit prior, leading to better generalisation in the small meta-dataset regime.

## 2.6 ATTENTION

One way to extend the BNNP would be to go beyond MLPs and incorporate a more sophisticated neural architecture, such as the attention layer (Vaswani et al., 2017). We outline two ways to do this. As is common in the NP literature, the encoder can be augmented with attention blocks (Kim et al., 2019; Nguyen and Grover, 2022). In the BNNP, this means parameterising the inference networks $\{g_{\theta_l}\}_{l=1}^L$ as transformers rather than MLPs, and processing the full context set together at each layer. This could improve performance since modelling interactions *between* context points could lead to better pseudo-likelihood parameter estimates than when we process each point independently. However, an attentive encoder incurs an extra computational cost of $\mathcal{O}(n_c^2)$. Furthermore, application of our within-task minibatching scheme under an attentive encoder would lead to different posteriors to the full-batch forward pass due to the new dependencies between context points. Nonetheless, we refer to this variant of the model as the attentive BNNP (AttBNNP).

Alternatively, we can focus on the decoder. The core technology that we introduce is a way to amortise inference over the weights of a linear layer situated arbitrarily within a neural network. Since an attention block is just a composition of linear layers and various nonparametric operations[3], it is possible to amortise inference in an attention block by amortising the inference in each linear layer therein. If such amortised attention blocks are stacked, we end up with a transformer whose weights posterior is obtained in-context. A more sophisticated decoder could enable us to model more complex tasks. However, such a decoder processes the target locations together such that the prediction for a particular target depends on the other target locations. This property destroys the consistency of the model and therefore no longer results in a valid stochastic process. We therefore refer to this version of the model as the *Bayesian neural attentive machine* (BNAM), omitting any reference to stochastic processes from its name. Note that the BNAM also incurs an extra $\mathcal{O}(n_t^2)$ computational cost. For computational diagrams as well as further details on the lack of consistency of the BNAM including a simple demo, see Appendix E.

## 3 RELATED WORK

**Neural Processes.** The BNNP is a new member of the NP family (Garnelo et al., 2018a; Dubois et al., 2020), and in particular the latent-variable NP family (Garnelo et al., 2018b; Singh et al., 2019; Lee et al., 2020; Foong et al., 2020). The AttBNNP is also a member of the transformer NP family (Kim et al., 2019; Nguyen and Grover, 2022; Feng et al., 2023; Ashman et al., 2024a;c). Our model is different to existing latent-variable NPs in that our latent variable is the parameterisation of the decoder itself, and not an abstract representation of the context set that is *passed to* the decoder. While Rochussen and Fortuin (2025) adopt a similar approach, in this work the decoder is a BNN rather than a sparse GP. Inference in the BNNP can be viewed as an instantiation of Volpp et al. (2021)'s Bayesian context aggregation mechanism—in both settings we have an encoder that maps from individual context points to pseudo-likelihood terms with which the latent variable's posterior is obtained through exact inference. While we choose to exclude the BNAM from the NP family due to its lack of consistency, some existing NP variants also lack consistency; Bruinsma et al. (2023)'s NP variants and Nguyen and Grover (2022)'s TNP-A both fail to produce consistent predictive distributions due to autoregression amongst targets, while Nguyen and Grover (2022)'s TNP-ND is

---

[3]Such as residual connections. We do not consider inference over the `layer-norm` parameters, justifying a deterministic treatment of them through their comparative inability to overfit.

more similar to the BNAM, with attention performed between targets[4]. Our training and modelling setup is also closely related to that of Gordon et al. (2019), with the first term of our proposed training objective being exactly equivalent to theirs (see Appendix A).

**Approximate Inference in BNNs.** There have been considerable efforts to develop more accurate and scalable approximate inference methods for BNNs in recent decades. There are Markov-chain Monte-Carlo algorithms (Neal, 1992; Welling and Teh, 2011; Sommer et al., 2024), ensemble and particle approaches (Lakshminarayanan et al., 2017; D' Angelo and Fortuin, 2021; Liu and Wang, 2016), Laplace approximations (MacKay, 1992; Ritter et al., 2018; Immer et al., 2021), variational strategies (Hinton and van Camp, 1993; Graves, 2011; Louizos and Welling, 2017), as well as more standalone solutions (Maddox et al., 2019; Gal and Ghahramani, 2016; Hernandez-Lobato and Adams, 2015). Our approach is most similar to Ober and Aitchison (2021)'s global inducing point variational posterior since we factorise the variational posterior into layerwise conditionals and compute each one through exact inference under pseudo-observations. Unlike their approach, in each layer we have a pseudo-likelihood corresponding to *every* datapoint rather than for a limited set of inducing locations; we parameterise the pseudo-likelihood terms via inference networks rather than directly; and we use a broader class of Gaussian priors. Another related method is Kurle et al. (2024)'s BALI, which adopts a non-variational approach to parameterising the pseudo-likelihood terms in each layer. Separately, our work shares the use of secondary/inference/"*hyper*" networks with normalising flow-based approaches (Louizos and Welling, 2017) and Bayesian hypernetworks (Krueger et al., 2018), but the difference is in what the secondary networks map between—for us it is from observations to pseudo-likelihood terms and for them it is from base distribution samples to posterior samples. Finally, our approximate inference scheme is an instance of structured variational inference (Hoffman and Blei, 2015).

**Priors in BNNs.** In general, there is no widely accepted way to select well-specified priors in BNNs (see Fortuin, 2022). While (meta-)learning priors is not a new concept (Rasmussen and Williams, 2006; Patacchiola et al., 2020; Fortuin et al., 2020; Rothfuss et al., 2021), it is a relatively under-explored approach in the context of Bayesian deep learning. To this end, Fortuin et al. (2022) analyse the empirical distributions of trained neural network weights to construct new priors, Shwartz-Ziv et al. (2022) use the approximate posterior from a large pre-training task as a learned prior for downstream fine-tuning tasks, and Villecroze et al. (2025) adopt an empirical variational Bayesian approach to learning a prior over the last-layer weights of a BNN. Ravi and Beatson (2019) and Rothfuss et al. (2021) consider meta-learning BNN priors, with the latter even doing so via AVI. However, they both do so within a hierarchical Bayesian framework involving restrictive hyper-priors. Concurrently with this work, Park et al. (2026) also devise a scheme for meta-learning BNN priors, but their weights posterior is obtained in-context via a network linearisation strategy rather than amortisation. They also propose a mixture of priors—our results suggest this is unnecessary.

## 4 EMPIRICAL INVESTIGATION

### 4.1 HOW GOOD IS THE BNNP'S APPROXIMATE POSTERIOR?

We begin by evaluating the quality of approximate inference in the BNNP. We adopt a similar setup to Bui (2021) and, under a common BNN architecture with fixed hyperparameters (including $\Psi$), we measure the gap between the log marginal likelihood (LML) and the ELBOs achieved by various VI techniques. The difference between these two quantities represents the KL divergence $\mathrm{KL}\big[q(\mathbf{W}|\mathcal{D}) \ \| \ p_\Psi(\mathbf{W}|\mathcal{D})\big]$, giving us a a clear metric of approximation quality in each case. To eliminate any bias in inference quality due to misspecified priors, we generate a dataset by 1.) sampling a function from the BNN's prior, 2.) uniformly sampling a set of inputs, and 3.) adding Gaussian noise with standard deviation $0.1$ to the outputs. Since the data therefore has a high probability of being generated under the prior (because it was), we can estimate the log marginal likelihood via Monte Carlo integration. We compare a BNNP meta-trained across similarly generated datasets under $\mathcal{L}_{\text{PP-AVI}}$, a BNNP trained on just the evaluation task via the standard ELBO objective function, mean-field VI (MFVI; Blundell et al., 2015), Ober and Aitchison (2021)'s global inducing-point VI (GIVI), as well as a number of VI algorithms with increasingly high-rank Gaussian variational posteriors: unitwise correlated (UCVI), layerwise-correlated (LCVI), and fully-correlated

---

[4]In their case, attention is performed between the target tokens only for the predictive covariance module. The predictive means for each target location remain conditionally independent given the context set.

(FCVI). The results, which are collected across a range of likelihood noise settings and averaged over four repeat runs, are shown in Fig. 2. They demonstrate that the BNNP is capable of high quality approximate inference. For all methods we see that approximate inference quality decreases with smaller likelihood noises. This effect is more pronounced for the unstructured variational approximations (MFVI, *CVI), so with these methods we observe the same bias in model selection via the ELBO as Bui (2021)—overly large $\sigma_y$'s are favoured (the true noise level is $10^{-1}$).

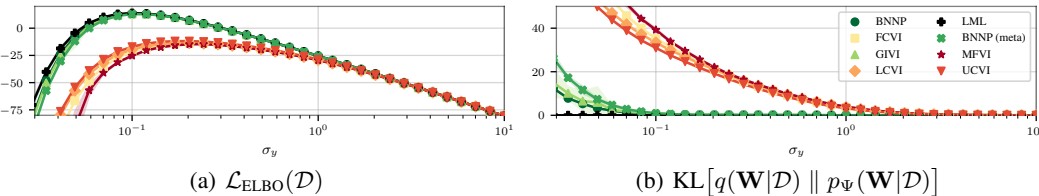

(a) $\mathcal{L}_{\text{ELBO}}(\mathcal{D})$        (b) $\text{KL}\big[q(\mathbf{W}|\mathcal{D}) \parallel p_\Psi(\mathbf{W}|\mathcal{D})\big]$

Figure 2: ELBO and KL divergence between approximate and true posteriors for different VI methods. The BNNP well-approximates the true posterior.

## 4.2 DO MEANINGFUL BNN PRIORS EXIST?

Here we train BNNPs (including $\Psi$) on meta-datasets with different data-generating processes. We then qualitatively compare functions sampled from the BNNP's learned prior with those from the true data-generating process. We consider synthetic meta-regression datasets generated from random sawtooth functions, Heaviside (/binary) functions, functions from a standard BNN prior[5], and synthetic ECG signals, as well as the MNIST dataset (LeCun et al., 1989) with random pixel masking cast as a pixelwise meta-regression dataset (Garnelo et al., 2018a), in which case we use the AttnBNNP. The results are visualised in Figs. 3 and 4.

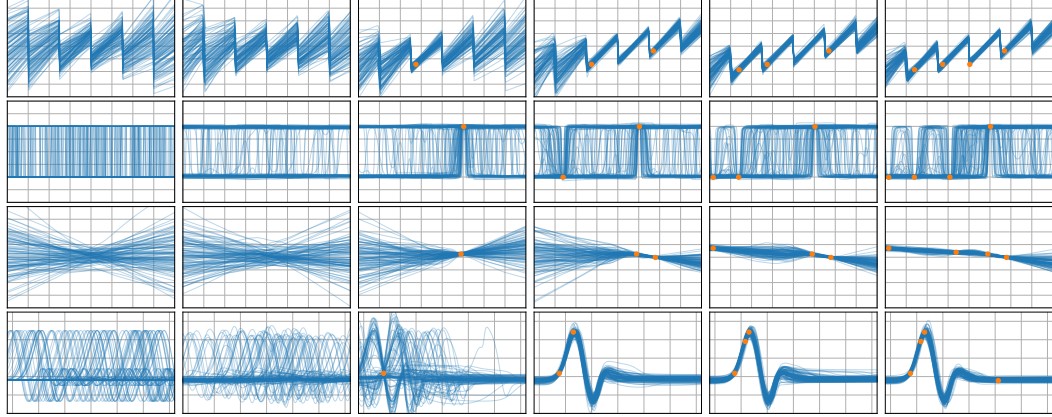

Figure 3: Function samples from the true data-generating process (first column), learned BNNP prior predictive samples (second column), BNNP posterior predictive function samples (remaining columns).

We observe that the BNNP learns functional priors that are almost indistinguishable from the synthetic data-generating processes. The corresponding posterior predictive samples appear to be sensible as well, with increased uncertainty away from observations while maintaining the underlying functional structure in each case. While the AttnBNNP's ability to generate MNIST images might not represent the state-of-the-art, the fact that many of the samples are clearly recognisable digits further demonstrates this section's conclusions, which are that 1) BNN priors that faithfully encode complex data-generating processes *do* exist, and 2) the BNNP can be used to find such priors.

## 4.3 DOES A GOOD PRIOR DOUBLE AS A LUNCH VOUCHER?

To investigate whether the quality of approximate inference still matters under a good prior, we consider one real-world and three synthetic data-generating processes. In each case, we meta-train

---

[5]Throughout our empirical investigation, we use *standard BNN prior* to refer to zero-centered fully-factorised Gaussian priors with variance scaled inversely proportional to layer width.

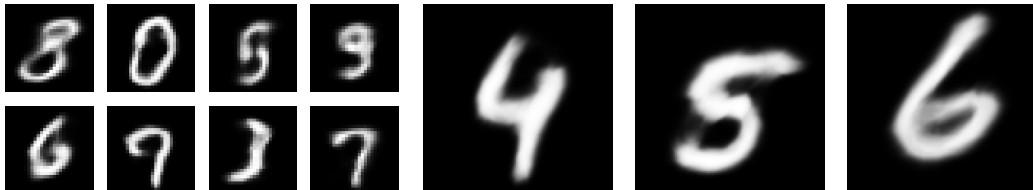

Figure 4: Generative modelling of MNIST digits with the AttnBNNP. Smaller images depict sampled prior functions evaluated at the same 28×28 grid of inputs that the training data lay on. Larger ones depict sampled functions queried at a 100×100 gid of inputs. The AttnBNNP's prior has encoded the functional behaviour of handwritten digits, so super-resolution is natively supported without needing further training.

a BNNP in order to find a well-specified BNN prior for the problem. For a variety of approximate inference algorithms, we compare predictive performance achieved under a standard prior versus a well-specified one. This is repeated over 16 test datasets and we use per-datapoint log posterior predictive density (LPPD) and mean absolute error (MAE) as metrics. We consider SWAG (Maddox et al., 2019), MFVI, Langevin Monte Carlo (LMC; Rossky et al., 1978), GIVI, the BNNP, and Hamiltonian Monte Carlo (HMC; Neal, 1992). For our real-world setting, we substitute LMC and HMC for their stochastic-gradient counterparts (SGLD; Welling and Teh, 2011, SGHMC; Chen et al., 2014) due to larger context sets and architectures.

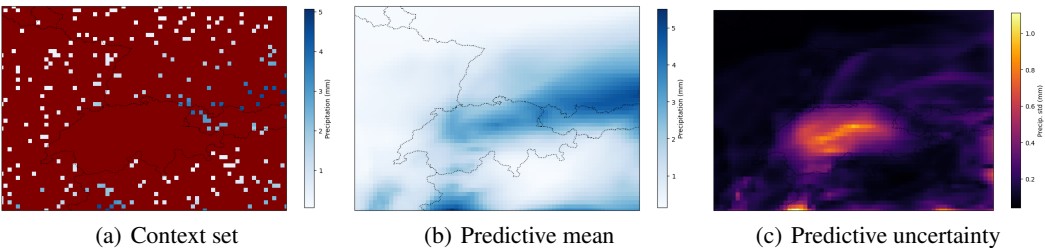

(a) Context set        (b) Predictive mean        (c) Predictive uncertainty

Figure 5: Demonstration of an ERA5 precipitation prediction test task with the BNNP. With no context points over Switzerland, the BNNP's predictive uncertainty is increased in that region.

As synthetic data-generating processes, we use squared-exponential GP, Heaviside, and sawtooth functions. In all three cases the inputs are sampled uniformly and the outputs are lightly noised. The real-world setting we consider is precipitation prediction over an area of Europe centered on Switzerland. We use the ERA5 Land dataset (Muñoz Sabater et al., 2021) and use longitude, latitude, and temperature as input variables. To make the prediction task even more challenging, we omit any context points from Switzerland in the test tasks, meaning predictions in this region are heavily influenced by the choice of prior. Fig. 5 demonstrates the ERA5 test task setup and Fig. 6 displays the results across the four settings. While it is clear that better priors boost predictive performance, we also see that there remains considerable variation in performance amongst the methods when under learned priors. In other words; a good prior is *not* all you need when working with BNNs since high quality approximate inference is still necessary—there is no free lunch in Bayesian deep learning.

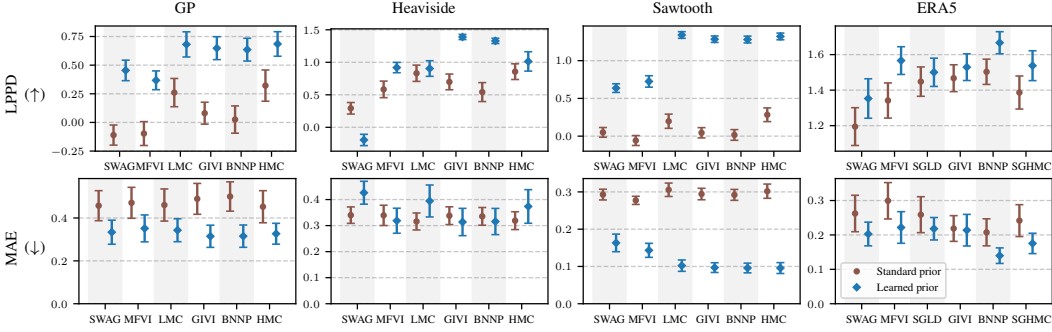

Figure 6: Target-set performance of approximate inference algorithms under a well-specified prior (blue) and a standard BNN prior (brown). The learned prior almost always leads to improved performance.

## 4.4 CAN A RESTRICTED PRIOR IMPROVE META-LEVEL DATA EFFICIENCY?

We consider two problem settings in which the number of available datasets would typically be seen as too few for NPs to be applied. The first setting is a recasting of the Abalone age prediction

task (Nash et al., 1994) as meta-regression. The three classes corresponding to a specimen's sex (male/female/infant) are used to separate the data into three distinct datasets. The male and female datasets are used for meta-training and the infant dataset is reserved for testing. There are seven input features including various specimen size and weight measurements. The second setting is based on the Paul15 single-cell RNA sequencing dataset (Paul et al., 2015), and the task is to predict how specialised a cell is from 3451 gene expressions. We perform PCA on the data to obtain 100 information-rich abstract features, and split the dataset into 19 subsets according to cell clusterings that Paul et al. (2015) provide. We randomly select ten for meta-training and one for testing.

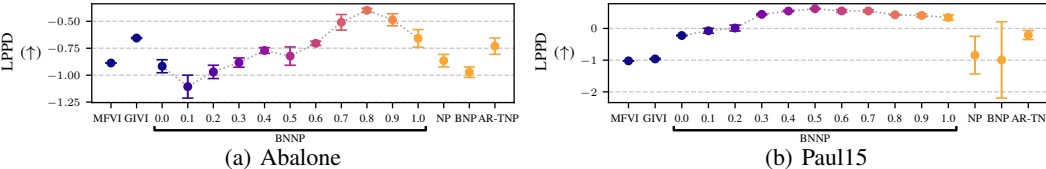

(a) Abalone                  (b) Paul15

Figure 7: Test-task target-set performance for two meta-learning problems with limited data. Decimals indicate the proportion of the BNNP's prior that is trained. BNNPs with partially trainable priors perform the best.

At the bottom end of the "prior-learnability" spectrum, we consider MFVI and GIVI baselines with standard priors (trained on the test-task's context set), while at the other, as comparable NPs which produce coherent function samples, we consider the original (latent-variable) NP (Garnelo et al., 2018b), the Bayesian NP (BNP; Volpp et al., 2021), and a transformer NP (Nguyen and Grover, 2022) under Bruinsma et al. (2023)'s state-of-the-art autoregressive sampling scheme (AR-TNP). We compare the baselines to the BNNP across its full range of prior-learnability where all non-learnable prior parameters are fixed to standard BNN prior settings, and prior parameters corresponding to earlier layer weights are made trainable first. The results are averaged over four trials. In Fig. 7 we see in both problem settings that the best-performing model is a BNNP with a *partially* learnable prior (0.8 and 0.5 respectively). In the particularly data-scarce Abalone problem, we see clear evidence of overfitting in the fully-flexible NPs, including the BNNP-1.0.

## 5 Discussion, Limitations, and the Bigger Picture

Why is the BNNP's approximate posterior as good as it is? Each layer's approximate posterior is explicitly conditioned on the weights of the previous layers, enabling us to model weight correlations *between* layers. While it is tempting to attribute the high quality to this as Ober and Aitchison (2021) do, FCVI's approximate posterior—which also models such inter-layer correlations—is very poor. One explanation could be that our method is unaffected by the posterior multimodality induced by weight-space symmetries. This is because each layer is given access to the outputs of the previous layer as well as its own pseudo-outputs, meaning each layer only "sees" one mode of the posterior.

While it may be surprising that simple Gaussian priors can yield highly complex, even multimodal (Heaviside in Fig. 3, MNIST digits in Fig. 4) stochastic process priors, we note that the universal posterior predictive approximation result for mean-field Gaussian posteriors of Farquhar et al. (2020) likely also applies to Gaussian priors. We are encouraged by the demonstrated flexibility of Gaussian priors as it means finding good BNN priors is not as insurmountable a task as it might have been. The BNNP provides a solution to this problem for the case when multiple datasets are available, but the problem remains unsolved for the single-dataset case.

We highlight that our message is *not* that the BNNP is the one model to rule them all. When there is a limited number of datasets, it is a very useful NP for practitioners to have in their inventory. Otherwise, our main focus in this work was in using it as a scientific tool with which to study BNNs. Inference in the BNNP scales unfavourably with architecture width (Appendix C), so we leave investigation of the BNNP's performance as a general-purpose NP to future work. Similarly, we introduce the BNAM for scientific interest but we do not include it in our experiments since it is of no use in answering our particular research questions.

Finally, we note that the strong performance of our explicitly Bayesian meta-learning setup hints at an underlying lesson; that Bayesians should be using the abundant data of today's world to learn powerful *priors*, not to be squeezing posteriors out of dubious priors. This message seems to be coming from an ever-growing chorus, with the posing of Bayesian inference as an in-context learning problem becoming increasingly popular (Müller et al., 2025; Reuter et al., 2025; Chang et al., 2025).

ACKNOWLEDGEMENTS

TR is grateful to Adrian Weller and Rich Turner for helpful comments and Matt Ashman for day-to-day supervision on an earlier version of this project (Ashman et al., 2023). VF was supported by the Branco Weiss Fellowship.

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

# A  OBJECTIVE FUNCTIONS AND TRAINING

In this section we justify our choice of objective function, we discuss practical implementation of it, and we discuss alternatives. Throughout, we use $q$ to denote distributions that depend on the approximate posterior $q(\mathbf{W}|\cdot)$ and $p_\Psi$ to denote distributions that depend on the parameters of the prior *without* depending on $q(\mathbf{W}|\cdot)$.

## A.1  PP-AVI

We begin by repeating Proposition 1 for the reader's benefit.

**Proposition 1.** *For $|\Xi| \to \infty$, maximisation of $\mathcal{L}_{PP\text{-}AVI}(\Xi)$ directly targets the three desiderata.*

In order to prove Proposition 1, we first provide formal definitions of the three desiderata. Throughout, we assume a true underlying data-generating process $\mathcal{D} \sim p(\mathcal{D})$.

**Definition 1** (accurate approximate posteriors)**.** *A probabilistic meta-learner produces accurate approximate posteriors if the task-averaged KL divergence between approximate and true posteriors*

$$\mathbb{E}_{p(\mathcal{D}_c)}\Big[KL\left[q(\mathbf{W}|\mathcal{D}_c) \parallel p_\Psi(\mathbf{W}|\mathcal{D}_c)\right]\Big] \tag{11}$$

*is small.*

**Definition 2** (faithful prior)**.** *A probabilistic meta-learner has a prior that faithfully encodes the data-generating process if the task-averaged KL divergence between the true generative process $p(\mathbf{Y}_c|\mathbf{X}_c)$ and the model's marginal likelihood $p_\Psi(\mathbf{Y}_c|\mathbf{X}_c)$,*

$$\mathbb{E}_{p(\mathbf{X}_c)}\Big[KL\big[p(\mathbf{Y}_c|\mathbf{X}_c) \parallel p_\Psi(\mathbf{Y}_c|\mathbf{X}_c)\big]\Big] \tag{12}$$

*is small.*

**Definition 3** (high quality predictions.)**.** *A probabilistic meta-learner produces high quality predictions if the task-averaged KL divergence between the true conditional generative process $p(\mathbf{Y}_t|\mathcal{D}_c, \mathbf{X}_t)$ and the model's posterior predictive $q(\mathbf{Y}_t|\mathcal{D}_c, \mathbf{X}_t)$,*

$$\mathbb{E}_{p(\mathcal{D}_c, \mathbf{X}_t)}\Big[KL\big[p(\mathbf{Y}_t|\mathcal{D}_c, \mathbf{X}_t) \parallel q(\mathbf{Y}_t|\mathcal{D}_c, \mathbf{X}_t)\big]\Big] \tag{13}$$

*is small.*

*Proof.* Recall the definition of $\mathcal{L}_{\text{PP-AVI}}(\Xi)$:

$$\mathcal{L}_{\text{PP-AVI}}(\Xi) := \frac{1}{|\Xi|}\sum_{j=1}^{|\Xi|} \log q\left(\mathbf{Y}_t^{(j)}|\mathcal{D}_c^{(j)}, \mathbf{X}_t^{(j)}\right) + \mathcal{L}_{\text{ELBO}}\left(\mathcal{D}_c^{(j)}\right). \tag{14}$$

Taking the limit of infinite datasets, we proceed as follows

$$\lim_{|\Xi| \to \infty} \mathcal{L}_{\text{PP-AVI}}(\Xi) = \mathbb{E}_{p(\mathcal{D})} \Big[ \log q \left( \mathbf{Y}_t | \mathcal{D}_c, \mathbf{X}_t \right) + \mathcal{L}_{\text{ELBO}} \left( \mathcal{D}_c \right) \Big] \tag{15}$$

$$= \mathbb{E}_{p(\mathcal{D})} \Big[ \log q \left( \mathbf{Y}_t | \mathcal{D}_c, \mathbf{X}_t \right) + \log p_\Psi (\mathbf{Y}_c | \mathbf{X}_c)$$
$$- \text{KL}[q(\mathbf{W}|\mathcal{D}_c) \, \| \, p_\Psi(\mathbf{W}|\mathcal{D}_c)] \Big] \tag{16}$$

$$= \color{blue}{\mathbb{E}_{p(\mathcal{D}_c, \mathbf{X}_t)} \Big[ \mathbb{E}_{p(\mathbf{Y}_t | \mathcal{D}_c, \mathbf{X}_t)} \left[ \log q(\mathbf{Y}_t | \mathcal{D}_c, \mathbf{X}_t) \right] \Big]}$$
$$\color{red}{+ \mathbb{E}_{p(\mathbf{X}_c)} \Big[ \mathbb{E}_{p(\mathbf{Y}_c | \mathbf{X}_c)} \left[ \log p_\Psi(\mathbf{Y}_c | \mathbf{X}_c) \right] \Big]}$$
$$- \mathbb{E}_{p(\mathcal{D}_c)} \Big[ \text{KL} \left[ q(\mathbf{W}|\mathcal{D}_c) \, \| \, p_\Psi(\mathbf{W}|\mathcal{D}_c) \right] \Big] \tag{17}$$

$$= \color{blue}{\mathbb{E}_{p(\mathcal{D}_c, \mathbf{X}_t)} \Big[ \mathbb{E}_{p(\mathbf{Y}_t | \mathcal{D}_c, \mathbf{X}_t)} \left[ \log \frac{q(\mathbf{Y}_t | \mathcal{D}_c, \mathbf{X}_t) p(\mathbf{Y}_t | \mathcal{D}_c, \mathbf{X}_t)}{p(\mathbf{Y}_t | \mathcal{D}_c, \mathbf{X}_t)} \right] \Big]}$$
$$\color{red}{+ \mathbb{E}_{p(\mathbf{X}_c)} \Big[ \mathbb{E}_{p(\mathbf{Y}_c | \mathbf{X}_c)} \left[ \log \frac{p_\Psi(\mathbf{Y}_c | \mathbf{X}_c) p \mathbf{Y}_c | \mathbf{X}_c)}{p(\mathbf{Y}_c | \mathbf{X}_c)} \right] \Big]}$$
$$- \mathbb{E}_{p(\mathcal{D}_c)} \Big[ \text{KL} \left[ q(\mathbf{W}|\mathcal{D}_c) \, \| \, p_\Psi(\mathbf{W}|\mathcal{D}_c) \right] \Big] \tag{18}$$

$$= \color{blue}{\underbrace{\mathbb{H}(\mathbf{Y}_t | \mathcal{D}_c, \mathbf{X}_t)}_{\text{const.}} - \mathbb{E}_{p(\mathcal{D}_c, \mathbf{X}_t)} \Big[ \text{KL} \big[ p(\mathbf{Y}_t | \mathcal{D}_c, \mathbf{X}_t) \, \| \, q(\mathbf{Y}_t | \mathcal{D}_c, \mathbf{X}_t) \big] \Big]}$$
$$\color{red}{+ \underbrace{\mathbb{H}(\mathbf{Y}_c | \mathbf{X}_c)}_{\text{const.}} - \mathbb{E}_{p(\mathbf{X}_c)} \Big[ \text{KL} \big[ p(\mathbf{Y}_c | \mathbf{X}_c) \, \| \, p_\Psi(\mathbf{Y}_c | \mathbf{X}_c) \big] \Big]}$$
$$- \mathbb{E}_{p(\mathcal{D}_c)} \Big[ \text{KL} \left[ q(\mathbf{W}|\mathcal{D}_c) \, \| \, p_\Psi(\mathbf{W}|\mathcal{D}_c) \right] \Big] \tag{19}$$

where $\mathbb{H}(\cdot|\cdot)$ denotes the Shannon conditional entropy. Since the entropy terms are constant with respect to the variational and model parameters $\{\Theta, \Psi\}$, we reach

$$\underset{\{\Theta, \Psi\}}{\arg\max} \lim_{|\Xi| \to \infty} \mathcal{L}_{\text{PP-AVI}}(\Xi) \equiv \underset{\{\Theta, \Psi\}}{\arg\min} \Big[ \mathbb{E}_{p(\mathcal{D}_c, \mathbf{X}_t)} \Big[ \text{KL} \big[ p(\mathbf{Y}_t | \mathcal{D}_c, \mathbf{X}_t) \, \| \, q(\mathbf{Y}_t | \mathcal{D}_c, \mathbf{X}_t) \big] \Big]$$
$$+ \mathbb{E}_{p(\mathbf{X}_c)} \Big[ \text{KL} \big[ p(\mathbf{Y}_c | \mathbf{X}_c) \, \| \, p_\Psi(\mathbf{Y}_c | \mathbf{X}_c) \big] \Big]$$
$$+ \mathbb{E}_{p(\mathcal{D}_c)} \Big[ \text{KL} \left[ q(\mathbf{W}|\mathcal{D}_c) \, \| \, p_\Psi(\mathbf{W}|\mathcal{D}_c) \right] \Big] \Big]. \tag{20}$$

$\square$

**Practical Details.** Unfortunately, both terms in $\mathcal{L}_{\text{PP-AVI}}$ are analytically intractable. As a first step, we decompose the ELBO into the usual *reconstruction error / complexity penalisation* form:

$$\mathcal{L}_{\text{PP-AVI}}(\Xi) = \frac{1}{|\Xi|} \sum_{j=1}^{|\Xi|} \log q \left( \mathbf{Y}_t^{(j)} | \mathcal{D}_c^{(j)}, \mathbf{X}_t^{(j)} \right) + \mathbb{E}_{q\left(\mathbf{W}|\mathcal{D}_c^{(j)}\right)} \Big[ \log p \left( \mathbf{Y}_c^{(j)} | \mathbf{W}, \mathbf{X}_c^{(j)} \right) \Big]$$
$$- \text{KL} \Big[ q \left( \mathbf{W}|\mathcal{D}_c^{(j)} \right) \, \| \, p_\Psi (\mathbf{W}) \Big]. \tag{21}$$

We deal with these three terms in order of increasing difficulty. The final term becomes tractable by decomposing it into the sum of layerwise KL divergences

$$\text{KL} \Big[ q \left( \mathbf{W}|\mathcal{D}_c^{(j)} \right) \, \| \, p_\Psi(\mathbf{W}) \Big] = \sum_{l=1}^{L} \text{KL} \Big[ q \left( \mathbf{W}^l | \mathbf{W}^{1:l-1}, \mathcal{D}_c^{(j)} \right) \, \| \, p_{\psi_l} \left( \mathbf{W}^l \right) \Big] \tag{22}$$

where each term in the sum is available in closed form as the KL divergence between two multivariate Gaussians.

The middle term, the (context set) *expected log-likelihood*, is unbiasedly estimable via Monte Carlo integration with a finite number of samples $K$

$$\mathbb{E}_{q\left(\mathbf{W}|\mathcal{D}_c^{(j)}\right)}\left[\log p\left(\mathbf{Y}_c^{(j)}|\mathbf{W}, \mathbf{X}_c^{(j)}\right)\right] \approx \frac{1}{K}\sum_{k=1}^{K}\log p\left(\mathbf{Y}_c^{(j)}|\mathbf{W}^{(k)}, \mathbf{X}_c^{(j)}\right) \tag{23}$$

where $\mathbf{W}^{(k)} \sim q\left(\mathbf{W}|\mathcal{D}_c^{(j)}\right)$ and low-variance gradient estimates are available via application of the reparameterisation trick (Kingma and Welling, 2014).

The posterior predictive term, which can equivalently be referred to as the (target set) *log expected likelihood*, is defined as

$$\log q\left(\mathbf{Y}_t^{(j)}|\mathcal{D}_c^{(j)}, \mathbf{X}_t^{(j)}\right) := \log \int p\left(\mathbf{Y}_t^{(j)}|\mathbf{W}, \mathbf{X}_t^{(j)}\right) q\left(\mathbf{W}|\mathcal{D}_c^{(j)}\right) \mathrm{d}\mathbf{W}. \tag{24}$$

Again turning to Monte Carlo integration with $K$ samples, we estimate this term via

$$\log q\left(\mathbf{Y}_t^{(j)}|\mathcal{D}_c^{(j)}, \mathbf{X}_t^{(j)}\right) \approx \log \frac{1}{K}\sum_{k=1}^{K} p\left(\mathbf{Y}_t^{(j)}|\mathbf{W}^{(k)}, \mathbf{X}_t^{(j)}\right) \tag{25}$$

$$= \mathrm{LogSumExp}\left(\left\{\log p\left(\mathbf{Y}_t^{(j)}|\mathbf{W}^{(k)}, \mathbf{X}_t^{(j)}\right)\right\}_{k=1}^{K}\right) - \log K \tag{26}$$

where $\mathbf{W}^{(k)} \sim q\left(\mathbf{W}|\mathcal{D}_c^{(j)}\right)$ and we use the log-sum-exp trick to maintain stable computations in log-space. Unfortunately, this Monte Carlo estimate is biased and so we cannot get away with single-sample estimates as we might with standard variational inference. In our experiments we used between 8 and 32 samples.

## A.2 STANDARD AVI / META EMPIRICAL BAYES

Perhaps the most obvious alternative objective function to consider would be the standard AVI objective given by the task-averaged ELBO:

$$\mathcal{L}_{\mathrm{AVI}}\left(\Xi\right) := \frac{1}{|\Xi|}\sum_{j=1}^{|\Xi|}\mathcal{L}_{\mathrm{ELBO}}\left(\mathcal{D}_c^{(j)}\right). \tag{27}$$

In the infinite-dataset limit, this objective function corresponds to minimising two of the three desired expected KL divergences; $\mathbb{E}_{p(\mathbf{X}_c)}\left[\mathrm{KL}\left[p(\mathbf{Y}_c|\mathbf{X}_c) \;\|\; p_\Psi(\mathbf{Y}_c|\mathbf{X}_c)\right]\right]$, and $\mathbb{E}_{p(\mathcal{D}_c)}\left[\mathrm{KL}\left[q(\mathbf{W}|\mathcal{D}_c) \| p_\Psi(\mathbf{W}|\mathcal{D}_c)\right]\right]$. Supposing we have globally optimised this objective function for an infinitely flexible model (not necessarily a BNNP) under infinite datasets, the trained model would have perfect approximate posteriors and a prior that models the true (context) data-generating process exactly. Perhaps the missing term would then be minimised for free—maybe we should expect extremely high quality posterior predictives from this resulting model. While this may be true (proper analysis left to future work), in practice we found $\mathcal{L}_{\mathrm{AVI}}$ to be significantly harder to globally optimise than $\mathcal{L}_{\mathrm{PP\text{-}AVI}}$. When training our BNNPs, we initialised the parameters of the prior to those of the standard isotropic Gaussian prior. For some tasks (particularly the heaviside and sawtooth function regression problems) it seems that this was a particularly adversarial initialisation in that maximisation of $\mathcal{L}_{\mathrm{AVI}}$ seemed not to be able to "break free" of the predictive behaviour induced by the standard BNN prior. So, even if the extra expected KL term from the PP-AVI objective is not strictly necessary, it seems to simplify the loss-landscape by suppressing the non-globally optimal modes via penalising miscalibrated posterior predictives.

Since the ELBO is a lower-bound to the log marginal likelihood $\mathcal{L}_{\mathrm{ELBO}}\left(\mathcal{D}_c\right) \leq \log p(\mathbf{Y}_c|\mathbf{X}_c)$, under infinite datasets we have that the AVI objective is a lower bound to the *expected* log marginal likelihood across tasks:

$$\lim_{|\Xi|\to\infty}\mathcal{L}_{\mathrm{AVI}}(\Xi) \leq \mathbb{E}_{p(\mathbf{X}_c)}\left[\log p(\mathbf{Y}_c|\mathbf{X}_c)\right]. \tag{28}$$

This then corresponds to a meta-level type-II marginal likelihood scheme, or, meta-level empirical Bayes. Note that this is the objective function used in Rochussen and Fortuin (2025) and Ashman et al. (2023).

### A.3 NP MAXIMUM LIKELIHOOD / POSTERIOR PREDICTIVE MAXIMISATION / ML-PIP

Neural processes tend to be used as probabilistic predictors more than anything else (such as generative models), so the only distribution users really care about is the posterior predictive $q(\mathbf{Y}_t|\mathcal{D}_c, \mathbf{X}_t)$. Note that while this distribution is referred to as the posterior predictive in the context of latent-variable NPs, for the conditional family of NPs it is referred to as the predictive likelihood. The obvious training scheme would then be to maximise the (log) posterior predictive likelihood of the target set over many tasks. This is the de-facto approach in conditional family NPs, and Gordon et al. (2019) demonstrated that it is sufficient for performing amortised variational inference in latent-variable meta-learners too. Gordon et al. refer to this scheme as *meta-learning probabilistic inference for prediction* (ML-PIP). We refer to this objective as the neural process maximum likelihood (NPML) objective, and it takes the form

$$\mathcal{L}_{\text{NPML}}(\Xi) := \frac{1}{|\Xi|} \sum_{j=1}^{|\Xi|} \log q(\mathbf{Y}_t^{(j)}|\mathcal{D}_c^{(j)}, \mathbf{X}_t^{(j)}). \tag{29}$$

In the infinite-dataset limit, maximising $\mathcal{L}_{\text{NPML}}$ corresponds to minimising the expected KL term that is missing from $\mathcal{L}_{\text{AVI}}$. In other words, we have that

$$\mathcal{L}_{\text{PP-AVI}}(\Xi) \equiv \mathcal{L}_{\text{NPML}}(\Xi) + \mathcal{L}_{\text{AVI}}(\Xi). \tag{30}$$

While NPML has been demonstrated to work well in NPs[6], with the BNNP we found it would lead either to solutions that modelled the data-generating process very badly, or to high numerical instability causing training runs to fail. As with $\mathcal{L}_{\text{AVI}}$, we suspect this is because of loss-landscape multimodality and difficulty ensuring global optimisation. In particular, we suspect that the globally optimal modes in $\mathcal{L}_{\text{AVI}}$ and $\mathcal{L}_{\text{NPML}}$ correspond to each other but that the secondary modes in each loss-landscape do not, meaning their sum is more straightforward to globally optimise.

### A.4 NEURAL PROCESS VARIATIONAL INFERENCE

Another way to train latent-variable NPs is through a more conventional variational approach, where the objective function is given by a lower bound to the conditional marginal likelihood $p_\Psi(\mathbf{Y}_t|\mathcal{D}_c, \mathbf{X}_t)$ averaged over tasks. We refer to this as the neural process variational inference (NPVI) objective, and it is defined as

$$\mathcal{L}_{\text{NPVI}}(\Xi) := \frac{1}{|\Xi|} \sum_{j=1}^{|\Xi|} \mathbb{E}_{q(\mathbf{W}|\mathcal{D}^{(j)})} \Big[ \log p\left(\mathbf{Y}_t^{(j)}|\mathbf{W}, \mathbf{X}_t^{(j)}\right) \Big]$$
$$- \text{KL}\Big[ q\left(\mathbf{W}|\mathcal{D}^{(j)}\right) \| p_\Psi\left(\mathbf{W}|\mathcal{D}_c^{(j)}\right) \Big]. \tag{31}$$

While NPVI is sensible in the infinite-dataset limit, being equivalent to minimising

$$\mathbb{E}_{p(\mathcal{D}_c, \mathbf{X}_t)} \Big[ \text{KL}\big[ p(\mathbf{Y}_t|\mathcal{D}_c, \mathbf{X}_t) \| p_\Psi(\mathbf{Y}_t|\mathcal{D}_c, \mathbf{X}_t) \big] \Big] + \mathbb{E}_{p(\mathcal{D})} \Big[ \text{KL}\big[ q(\mathbf{W}|\mathcal{D}) \| p_\Psi(\mathbf{W}|\mathcal{D}) \big] \Big], \tag{32}$$

it is intractable due to the presence of the true posterior $p_\Psi(\mathbf{W}|\mathcal{D}_c)$. While this is typically circumvented by approximating the true posterior with the approximate posterior $q(\mathbf{W}|\mathcal{D}_c)$, NPVI only targets Eq. (32) if the approximate posterior is exact. Under a poor approximate posterior, such as at the beginning of training, the loss-landscape is then quite different to what it should be and it is unclear how close the implemented version of $\mathcal{L}_{\text{NPVI}}$ ever becomes to the ideal one. Indeed, the NPVI approach is known to yield suboptimal predictive performance (Le et al., 2018). Furthermore, the KL term involves computing the two approximate posteriors $q(\mathbf{W}|\mathcal{D})$ and $q(\mathbf{W}|\mathcal{D}_c)$. Since the computational bottleneck associated with the BNNP is in computing the approximate posterior, the double approximate posterior computation renders $\mathcal{L}_{\text{NPVI}}$ even more inappropriate for the BNNP, and so we did not attempt to use it at all. Note that $\mathcal{L}_{\text{NPVI}}$ was the objective function used in the original (latent-variable) neural process (Garnelo et al., 2018b).

---

[6]Note that, for latent-variable type NPs, the latent variable is denoted by $\mathbf{z}$ rather than $\mathbf{W}$.

## A.5 TELL-AVI

Motivated to find an objective function that can be ubiasedly estimated and which has similar infinite-dataset behaviour to $\mathcal{L}_{\text{PP-AVI}}$, we introduce the target-expected-log-likelihood (TELL) AVI objective. We define it as

$$\mathcal{L}_{\text{TELL-AVI}}(\Xi) := \frac{1}{|\Xi|} \sum_{j=1}^{|\Xi|} \mathbb{E}_{q\left(\mathbf{W}|\mathcal{D}_c^{(j)}\right)} \left[\log p\left(\mathbf{Y}_t^{(j)}|\mathbf{W}, \mathbf{X}_t^{(j)}\right)\right] + \mathcal{L}_{\text{ELBO}}\left(\mathcal{D}_c^{(j)}\right) \tag{33}$$

where we note the only difference between $\mathcal{L}_{\text{PP-AVI}}$ and $\mathcal{L}_{\text{TELL-AVI}}$ is in the ordering of the $\log$ and expectation $\mathbb{E}_{q\left(\mathbf{W}|\mathcal{D}_c^{(j)}\right)}$ in the left-hand term[7]. The right-hand term is estimated exactly as in $\mathcal{L}_{\text{PP-AVI}}$, but the left-hand can be *unbiasedly* estimated by its Monte Carlo estimate from $K$ samples, $\frac{1}{K} \sum_{k=1}^{K} \log p\left(\mathbf{Y}_t^{(j)}|\mathbf{W}^{(k)}, \mathbf{X}_t^{(j)}\right)$ for $\mathbf{W}^{(k)} \sim q\left(\mathbf{W}|\mathcal{D}_c^{(j)}\right)$, enabling the use of very few samples for decreased computational cost.

Seeking to rewrite the expected log-likelihood term in terms of the log posterior predictive (log expected likelihood), we have

$$\mathbb{E}_{q(\mathbf{W}|\mathcal{D}_c)} \left[\log p\left(\mathbf{Y}_t|\mathbf{W}, \mathbf{X}_t\right)\right] = \mathbb{E}_{q(\mathbf{W}|\mathcal{D}_c)} \left[\log \frac{p\left(\mathbf{Y}_t|\mathbf{W}, \mathbf{X}_t\right) q(\mathbf{W}|\mathcal{D}_c) q\left(\mathbf{Y}_t|\mathcal{D}_c, \mathbf{X}_t\right)}{q(\mathbf{W}|\mathcal{D}_c) q\left(\mathbf{Y}_t|\mathcal{D}_c, \mathbf{X}_t\right)}\right] \tag{34}$$

$$= \mathbb{E}_{q(\mathbf{W}|\mathcal{D}_c)} \left[\log q\left(\mathbf{Y}_t|\mathcal{D}_c, \mathbf{X}_t\right)\right] + \mathbb{E}_{q(\mathbf{W}|\mathcal{D}_c)} \left[\log \frac{q\left(\mathbf{Y}_t, \mathbf{W}|\mathcal{D}_c, \mathbf{X}_t\right)}{q(\mathbf{W}|\mathcal{D}_c) q\left(\mathbf{Y}_t|\mathcal{D}_c, \mathbf{X}_t\right)}\right] \tag{35}$$

$$= \log q\left(\mathbf{Y}_t|\mathcal{D}_c, \mathbf{X}_t\right) + \mathbb{E}_{q(\mathbf{W}|\mathcal{D}_c)} \left[\log \frac{q\left(\mathbf{W}|\mathbf{X}_t, \mathbf{Y}_t, \mathcal{D}_c\right)}{q(\mathbf{W}|\mathcal{D}_c)}\right] \tag{36}$$

$$= \log q\left(\mathbf{Y}_t|\mathcal{D}_c, \mathbf{X}_t\right) - \text{KL}\left[q(\mathbf{W}|\mathcal{D}_c) \| q(\mathbf{W}|\mathcal{D})\right], \tag{37}$$

which gives us the following relationship between $\mathcal{L}_{\text{TELL-AVI}}$ and $\mathcal{L}_{\text{PP-AVI}}$:

$$\lim_{|\Xi|\to\infty} \mathcal{L}_{\text{TELL-AVI}} = \lim_{|\Xi|\to\infty} \mathcal{L}_{\text{PP-AVI}} - \mathbb{E}_{p(\mathcal{D})}\left[\text{KL}\left[q(\mathbf{W}|\mathcal{D}_c) \| q(\mathbf{W}|\mathcal{D})\right]\right]. \tag{38}$$

Though this expected KL term is somewhat sensible when interpreted as encouraging approximate posteriors from partial datasets to be similar to their full-data counterparts, the reverse interpretation (that it encourages full-data approximate posteriors to be only as good as their partial-data counterparts) highlights its dubiousness. Furthermore, the extra term would serve as a distractor from the more important "posteriors KL term" that $\mathcal{L}_{\text{PP-AVI}}$ already involves; $\mathbb{E}_{q(\mathbf{W}|\mathcal{D}_c)}\left[\text{KL}\left[q(\mathbf{W}|\mathcal{D}_c) \| p_\Psi(\mathbf{W}|\mathcal{D}_c)\right]\right]$, and it is unclear what behaviour the sum of the two KL terms involving $q(\mathbf{W}|\mathcal{D}_c)$ leads to. In preliminary experiments we found that $\mathcal{L}_{\text{TELL-AVI}}$ led to reasonable performance in terms of predictions and the resulting learned prior, but it was never quite as good as $\mathcal{L}_{\text{PP-AVI}}$.

## A.6 TRAINING

---

**Algorithm 1** BNNP training algorithm.

---

**Require:** meta-dataset $\Xi$, number of training steps $T$, number of Monte Carlo samples $k$,
    **for** $t = 1, \ldots, T$ **do**
        Randomly select $\xi$ from $\Xi$         ▷ select minibatch of tasks
        $\mathcal{L}_{\text{batch}} \leftarrow 0.0$         ▷ initialise batch objective
        **for** dataset $\mathcal{D}$ in minibatch **do**
            compute layerwise posteriors, obtain $k$ samples from each     ▷ e.g. by using algorithm 2
            compute $\mathcal{L}_{\text{PP-AVI}}(\mathcal{D})$ using $k$ samples     ▷ e.g. using practical details from section A.1
            $\mathcal{L}_{\text{batch}} \leftarrow \mathcal{L}_{\text{batch}} + \frac{1}{|\xi|} \mathcal{L}_{\text{PP-AVI}}(\mathcal{D})$
        **end for**
        Use $-\nabla \mathcal{L}_{\text{batch}}$ to update $\Theta$ and $\Psi$         ▷ e.g. using Adam, gradient descent, ...
    **end for**

---

[7]i.e., $\mathcal{L}_{\text{PP-AVI}}$ might have equivalently been called the target-log-expected-likelihood (TLEL) AVI objective.

## B    INFERENCE WITH ALTERNATIVE PRIORS

**Fully factorised.** The simplest form of prior is a fully factorised one $p_{\psi_l}(\mathbf{W}^l) =$ $\prod_{d=1}^{d_l} \prod_{d'=1}^{d_{l-1}} \mathcal{N}\left(w_{d,d'}^l; \mu_{d,d'}^l, \sigma_{d,d'}^l{}^2\right)$. Inference with this prior is performed similarly to the unit-wise factorised prior, except that the unitwise covariance matrix is given by $\mathbf{\Sigma}_d^l = \mathrm{diag}\left(\boldsymbol{\sigma}_d^l{}^2\right)$ and can therefore be inverted by taking the reciprocal of the diagonal elements, taking $\mathcal{O}(d_{l-1})$ time rather than $\mathcal{O}(d_{l-1}{}^3)$ time.

**Layerwise matrix-Gaussian.**    While a seemingly obvious prior to consider is the matrix-Gaussian over layerwise weight matrices (Kurle et al., 2024; Louizos and Welling, 2016; Ritter et al., 2018), this prior turns out not to fit nicely into our amortisation framework. The pseudo-observation noise variances in each layer would have to become part of the prior rather than the pseudo likelihoods, meaning our inference networks would lose their ability to up- or down-weight pseudo-observations by predicting their corresponding noise level. In other words, using matrix-Gaussian priors would force us to destroy the Bayesian context aggregation behaviour of our method.

**Layerwise full-rank Gaussian.**    A richer prior than the unitwise factorised one is a layerwise full-rank Gaussian defined over $\mathrm{vec}(\mathbf{W}^l)$. To lighten the notation, we define $\boldsymbol{\omega}^l := \mathrm{vec}(\mathbf{W}^l)$. The prior is then $p_{\psi_l}(\mathbf{W}^l) = p(\boldsymbol{\omega}^l) = \mathcal{N}\left(\boldsymbol{\omega}^l; \boldsymbol{\mu}^l, \mathbf{\Sigma}^l\right)$. In order to apply the standard (single-output) Bayesian linear regression results, we need to augment the data matrices $\mathbf{X}^{l-1}, \mathbf{Y}^l$ to some matrix-vector pair $\boldsymbol{\chi}_\phi^{l-1}, \boldsymbol{y}^l$, where $\boldsymbol{y}^l := \mathrm{vec}(\mathbf{Y}^l)$, so that the likelihood model can be written as $\boldsymbol{y}^l = \boldsymbol{\chi}_\phi^{l-1} \boldsymbol{\omega}^l$. It turns out that the "vec trick" (Petersen and Pedersen, 2008) of the Kronecker product $\otimes$ gives us what we need:

$$\mathbf{Y}^l = \phi(\mathbf{X}^{l-1})\mathbf{W}^l \tag{39}$$

$$\therefore \tag{40}$$

$$\boldsymbol{y}^l = \mathrm{vec}\left(\phi(\mathbf{X}^{l-1})\mathbf{W}^l\right) \tag{41}$$

$$= \mathrm{vec}\left(\phi(\mathbf{X}^{l-1})\mathbf{W}^l\mathbf{I}^\top\right) \tag{42}$$

$$= \left(\mathbf{I} \otimes \phi(\mathbf{X}^{l-1})\right)\boldsymbol{\omega}^l \tag{43}$$

giving us that $\boldsymbol{\chi}_\phi^{l-1} := \mathbf{I} \otimes \phi(\mathbf{X}^{l-1})$. Note that $\boldsymbol{\chi}_\phi^{l-1} \in \mathbb{R}^{d_l N \times d_l d_{l-1}}$. The posterior is therefore

$$p\left(\boldsymbol{\omega}^l | \mathbf{X}^{l-1}, \mathbf{Y}^l\right) = \mathcal{N}\left(\boldsymbol{\omega}^l; \mathbf{m}^l, \mathbf{S}^l\right) \tag{44}$$

with mean vector and covariance matrix given by

$$\mathbf{S}^{l-1} = \mathbf{\Sigma}^{l-1} + \boldsymbol{\chi}_\phi^{l-1}{}^\top \mathbf{\Lambda}^l \boldsymbol{\chi}_\phi^{l-1} \tag{45}$$

$$\mathbf{m}^l = \mathbf{S}^l\left(\mathbf{\Sigma}^{l-1}\boldsymbol{\mu}^l + \boldsymbol{\chi}_\phi^{l-1}{}^\top \mathbf{\Lambda}^l \boldsymbol{y}^l\right) \tag{46}$$

where $\mathbf{\Lambda}^l \in \mathbb{R}^{Nd_l \times Nd_l}$ is a diagonal precision matrix formed by stacking the $\mathbf{\Lambda}_d^l$ matrices along the leading diagonal of an $Nd_l \times Nd_l$ zeroes matrix. More specifically, $\mathbf{\Lambda}^l := \mathrm{diag}\left(\mathrm{concat}\left(\{\boldsymbol{\lambda}_d^l\}_{d=1}^{d_l}\right)\right)$ where the $n$-th element of $\boldsymbol{\lambda}_d^l \in \mathbb{R}^N$ is given by $\frac{1}{\sigma_{n,d}^l{}^2}$. To avoid direct matrix multiplications with the $d_l N \times d_l d_{l-1}$ matrix $\boldsymbol{\chi}_\phi^{l-1}$, Eq. (45) and Eq. (46) can be re-written as

$$\mathbf{S}^{l-1} = \mathbf{\Sigma}^{l-1} + \mathbf{\Phi}^l \tag{47}$$

$$\mathbf{m}^l = \mathbf{S}^l\left(\mathbf{\Sigma}^{l-1}\boldsymbol{\mu}^l + \boldsymbol{\phi}^l\right) \tag{48}$$

where $\mathbf{\Phi}^l$ is constructed by stacking the $d_{l-1} \times d_{l-1}$ matrices $\{\phi(\mathbf{X}^{l-1})^\top \mathbf{\Lambda}_d^l \phi(\mathbf{X}^{l-1})\}_{d=1}^{d_l}$ along the leading diagonal of a $d_l d_{l-1} \times d_l d_{l-1}$ zeroes matrix, and $\boldsymbol{\phi}^l$ is given by $\mathrm{vec}\left(\phi(\mathbf{X})^{l-1}{}^\top \tilde{\mathbf{Y}}^l\right)$ where the elements of $\tilde{\mathbf{Y}}^l$ are given by $\tilde{y}_{n,d}^l := \frac{y_{n,d}^l}{\sigma_{n,d}^l{}^2}$.

Note that inference with this type of prior is significantly more expensive than with the unitwise or weightwise factorised priors. This is because the computation is dominated by the inversion of $d_l d_{l-1} \times d_l d_{l-1}$ covariance matrices, requiring $\mathcal{O}\left(d_l^3 d_{l-1}^3\right)$ operations. For a network with uniform hidden layer width, the cost of inference therefore scales sextically $\left(d^6\right)$ with network width.

**Full-rank Gaussian.** The richest prior we consider is a full-rank Gaussian defined over all network weights $\boldsymbol{\omega} := \text{concat}\left(\{\text{vec}(\mathbf{W}^l)\}_{l=1}^L\right)$, i.e., $p(\mathbf{W}^{1:L}) = p(\boldsymbol{\omega}) = \mathcal{N}\left(\boldsymbol{\omega}; \boldsymbol{\mu}, \boldsymbol{\Sigma}\right)$. Inference is then performed in the same way as with the layerwise full-rank Gaussian prior, save that the layerwise factors $\{p_{\psi_l}(\mathbf{W}^l)\}_{l=1}^L$ are replaced with the layerwise conditionals $\{p(\mathbf{W}^l|\mathbf{W}^{1:l-1})\}_{l=1}^L$. Each layerwise conditional takes the form

$$p(\mathbf{W}^l|\mathbf{W}^{1:l-1}) = \int p(\boldsymbol{\omega}^{l:L}|\boldsymbol{\omega}^{1:l-1})\mathrm{d}\boldsymbol{\omega}^{l+1:L} = \mathcal{N}\left(\boldsymbol{\omega}^l; \boldsymbol{\mu}^{c|p}, \boldsymbol{\Sigma}^{c|p}\right) \tag{49}$$

where $c$ and $p$ denote the indices of the **c**urrent and **p**revious layer weights respectively, and the conditional mean and covariance are given by

$$\boldsymbol{\mu}^{c|p} = \boldsymbol{\mu}_c + \boldsymbol{\Sigma}_{cp}\boldsymbol{\Sigma}_{pp}^{-1}\left(\boldsymbol{\omega}^{1:l-1} - \boldsymbol{\mu}_p\right) \tag{50}$$

$$\boldsymbol{\Sigma}^{c|p} = \boldsymbol{\Sigma}_{cc} - \boldsymbol{\Sigma}_{cp}\boldsymbol{\Sigma}_{pp}^{-1}\boldsymbol{\Sigma}_{pc} \tag{51}$$

where $\boldsymbol{\omega}^a$ denotes the (vectorised) weights of layer $a$, $\boldsymbol{\mu}_a$ denotes the subvector of $\boldsymbol{\mu}$ indexed by $a$, and similarly $\boldsymbol{\Sigma}_{ab}$ denotes the submatrix of $\boldsymbol{\Sigma}$ indexed by $a$ and $b$.

# C   MINIBATCHING

Our minibatched posterior sampling algorithm for scalable inference in the BNNP is detailed in Algorithm 2. The algorithm describes the usual full-batch posterior sampling procedure if the number of minibatches is set to $B = 1$. We use `partial_forward_pass(`$\mathbf{X}, \mathbf{W}^{1:l}$`)` to refer to the propagation of inputs $\mathbf{X}$ forward to the $l$-th layer of the network using weights $\mathbf{W}^{1:l}$, and `compute_posterior` to refer to computation of Eq. (5).

---

**Algorithm 2** A minibatched BNNP posterior sampling algorithm.

---

**Require:** inference networks $\{g_{\theta_l}\}_{l=1}^L$ and priors $\{p_{\psi_l}(\mathbf{W}^l)\}_{l=1}^L$,
  **for** $l = 1, \ldots, L$ **do**
    **for** $b = 1, \ldots, B$ **do**
      Obtain $\mathcal{D}_b$
      **if** $l = 1$ **then**
        $\mathbf{X}_b^0 \leftarrow \mathbf{X}_b$
      **else**
        $\mathbf{X}_b^{l-1} \leftarrow$ `partial_forward_pass(`$\mathbf{X}_b, \mathbf{W}_{1:B}^{1:l-1}$`)`
      **end if**
      **if** $b = 1$ **then**
        $q(\mathbf{W}^l|\mathbf{W}_{1:B}^{1:l-1}, \mathcal{D}_1) \leftarrow$ `compute_posterior`$\big(\mathbf{X}_1^{l-1}, \mathcal{D}_1, g_{\theta_l}, p_{\psi_l}(\mathbf{W}^l)\big)$
      **else**
        $q(\mathbf{W}^l|\mathbf{W}_{1:B}^{1:l-1}, \mathcal{D}_{1:b}) \leftarrow$ `compute_posterior`$\big(\mathbf{X}_b^{l-1}, \mathcal{D}_b, g_{\theta_l}, q(\mathbf{W}^l|\mathbf{W}_{1:B}^{1:l-1}, \mathcal{D}_{1:b-1})\big)$
      **end if**
      Discard $\mathcal{D}_b$
    **end for**
    Sample $\mathbf{W}_{1:B}^l \sim q(\mathbf{W}^l|\mathbf{W}_{1:B}^{1:l-1}, \mathcal{D}_{1:B})$
  **end for**
  **return** $\{\mathbf{W}_{1:B}^l\}_{l=1}^L$

---

The time complexity associated with a forward pass through the $l$-th layer is $\mathcal{O}(n_c {d_{l-1}}^3 d_l)$, and the corresponding space complexity is $\mathcal{O}(n_c {d_{l-1}}^2 d_l)$. The superlinear scaling with the number of input neurons $d_{l-1}$ arises from inverting $d_{l-1} \times d_{l-1}$ matrices. The purpose of our minibatching algorithm is to reduce the linear scaling of the space complexity with context size to constant, i.e. to convert the space complexity to $\mathcal{O}(|b| {d_{l-1}}^2 d_l)$, where $|b|$ is the batch size.

While this is greatly beneficial at prediction time, during training it is necessary to store in memory the gradient information corresponding to all batches. This reverts the memory complexity to $\mathcal{O}(n_c {d_{l-1}}^2 d_l)$ in spite of the minibatching procedure. To remedy this, we propose to randomly select just one of the minibatches to compute gradients with respect to at every layer (i.e. the same minibatch for all layers), discarding gradient information for the other minibatches. While this leads to noisier gradient update steps, it reduces the memory complexity back down to $\mathcal{O}(|b| {d_{l-1}}^2 d_l)$ as desired. Although we leave a detailed analysis of any biases that this might introduce to future work, we emphasise here that we expect this scheme to maintain a distinct advantage over a naive minibatching in which we simply limit the maximum context set size to the batch size—our scheme should preserve the BNNP's ability to generalise predictive inference to context set sizes *larger than the batch size*.

# D    ONLINE LEARNING VIA LAST-LAYER SEQUENTIAL BAYESIAN INFERENCE

As mentioned in Section 2.4, any form of posterior update due to new data must be applied in full to each layer's posterior before computing the next layer's conditional posterior. Unfortunately, this means that sequential Bayesian inference cannot be used naïvely for online learning in the BNNP. To see why, consider a partioning of a task's dataset into "original" and "update" subsets $\mathcal{D} = \mathcal{D}_o \cup \mathcal{D}_u$. Assume we have already computed the layerwise posteriors for the original data $\{q(\mathbf{W}^l|\mathbf{W}_o^{1:l-1}, \mathcal{D}_o)\}_{l=1}^L$, and have since discarded the original data. Without access to the original data we can only obtain the conditional posteriors $\{q(\mathbf{W}^l|\mathbf{W}_o^{1:l-1}, \mathcal{D})\}_{l=1}^L$, and not what we need, which is $\{q(\mathbf{W}^l|\mathbf{W}_{o,u}^{1:l-1}, \mathcal{D})\}_{l=1}^L$. This is because conditioning on samples $\mathbf{W}_{o,u}^{1:l-1}$ requires propagating the full collection of data.

However, we can approximate what we need by updating just the last layer's posterior while preserving the existing previous layer posterior samples, giving us

$$q(\mathbf{W}|\mathcal{D}) \approx q(\mathbf{W}^L|\mathbf{W}_o^{1:l-1}, \mathcal{D}) \prod_{l=1}^{L-1} q(\mathbf{W}^l|\mathbf{W}_o^{1:l-1}, \mathcal{D}_o) \tag{52}$$

$$\propto p\left(\mathbf{Y}_u^L|\mathbf{W}^L, \mathbf{X}_u^{L-1}\right) q(\mathbf{W}^L|\mathbf{W}_o^{1:L-1}, \mathcal{D}_o) \prod_{l=1}^{L-1} q(\mathbf{W}^l|\mathbf{W}_o^{1:l-1}, \mathcal{D}_o) \tag{53}$$

We can interpret this as the first $L-1$ layers being a feature selector whose weights have been inferred from just the original data, while the weights of the prediction head (last layer) are updated in light of the new data by sequential Bayesian inference. Although this might seem like a tenuous approximation, we find it works well in practice. We found that the approximation deteriorates for increasingly deep architectures—this is unsurprising given that it corresponds to the posteriors of an increasingly small proportion of weights getting updated.

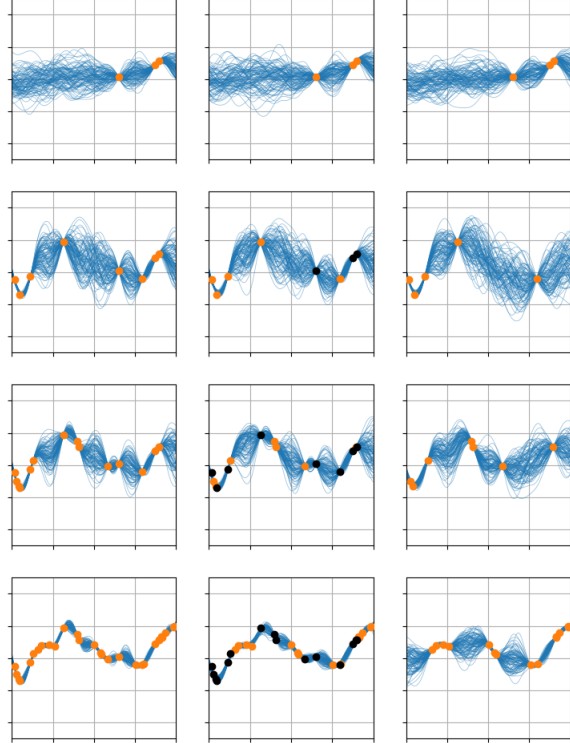

Figure 8: A demo of our online learning algorithm for the BNNP. Orange dots are context points, blue lines are posterior predictive samples, black dots are context points that we have lost access to. Each row incorporates more context data. The left-hand column corresponds to a BNNP given access to the *full* context set at each increment, the central column corresponds to our online learning algorithm, and the right-hand column corresponds to the BNNP given *only* the new context data.

# E   THE BAYESIAN NEURAL ATTENTIVE MACHINE

## E.1   ARCHITECTURE

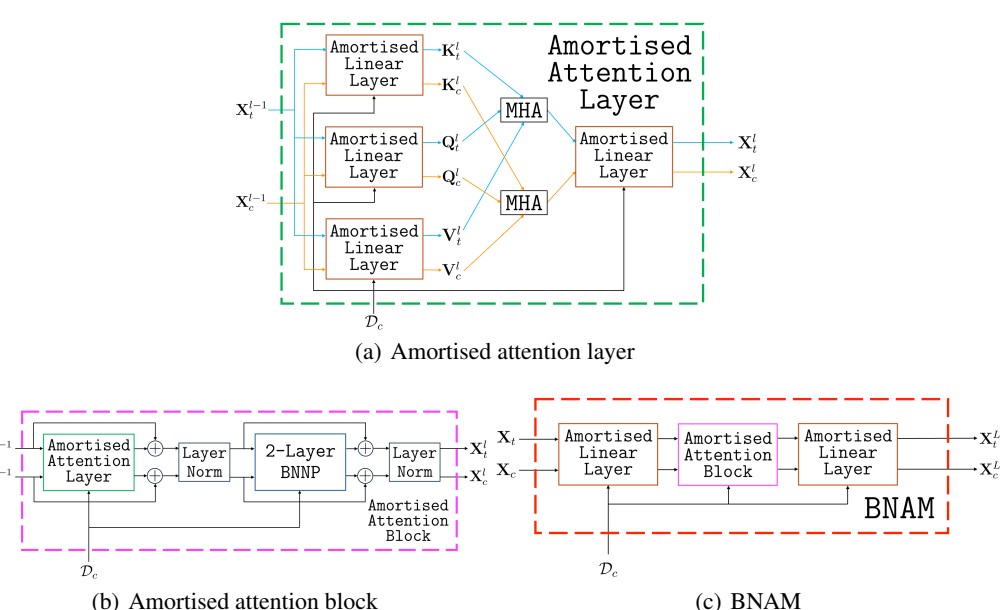

(a) Amortised attention layer

(b) Amortised attention block

(c) BNAM

Figure 9: Computational diagrams of the amortised attention layer (a), amortised attention block (b), and BNAM (c). Due to the numerous crossing lines in (a), we colour code the context and target input data paths as orange and light blue respectively. Arbitrarily many amortised attention blocks can be stacked sequentially in the BNAM; our diagram shows the simplest possible BNAM architecture.

We see in Fig. 9(a) that amortised inference can be performed in an attention layer by using amortised linear layers in place of standard linear layers, where MHA is the usual multi-head dot-product attention mechanism acting on keys $\mathbf{K}$, queries $\mathbf{Q}$, and values $\mathbf{V}$. Similarly, in Fig. 9(b) we follow the standard approach (Vaswani et al., 2017) for constructing stackable attention *blocks* from attention layers, residual connections, layer norms, and 2-layer MLPs, but replacing each of the attention layer and MLP with their amortised counterparts. In Fig. 9(c) we show how amortised inference can be performed in a transformer by composing amortised linear layers and amortised attention blocks. We note that the resulting model can only be used in a somewhat unusual way for transformers; to map from test inputs $\mathbf{X}_t$ to predicted test outputs $\mathbf{Y}_t$ where attention is performed between the *test* inputs, and where the posterior over the transformer's weights is estimated from a context set.

## E.2   LACK OF CONSISTENCY

As mentioned in the main text, the BNAM does not produce consistent predictive distributions over target outputs. As we shall see, it is the attention *between test inputs* that causes this. For any finite set of target locations $\mathbf{X}_{t_{1:n}}$, the joint distributions $p_{\mathbf{X}_{t_{1:n}}}(\mathbf{Y}_{t_{1:n}})$ and $p_{\mathbf{X}_{t_{1:m}}}(\mathbf{Y}_{t_{1:m}})$ for $m < n$ are (marginalisation) *consistent* if

$$p_{\mathbf{X}_{t_{1:m}}}(\mathbf{Y}_{t_{1:m}}) = \int p_{\mathbf{X}_{t_{1:n}}}(\mathbf{Y}_{t_{1:n}}) \mathrm{d}\mathbf{Y}_{t_{m+1:n}}. \tag{54}$$

In the case of the BNAM, we are interested in joint distributions of the form

$$p\left(\mathbf{Y}_{t_{1:n}}|\mathcal{D}_c, \mathbf{X}_{t_{1:n}}\right) = \int p\left(\mathbf{Y}_{t_{1:n}}|\mathbf{W}, \mathbf{X}_{t_{1:n}}\right) p_\Psi\left(\mathbf{W}|\mathcal{D}_c\right) \mathrm{d}\mathbf{W} \tag{55}$$

where the weights posterior $p_\Psi(\mathbf{W}|\mathcal{D}_c)$ is approximated by the variational posterior $q(\mathbf{W}|\mathcal{D}_c) = \prod_{l=1}^{L} q(\mathbf{W}^l|\mathbf{W}^{1:l-1}, \mathcal{D}_c)$ that we developed in the paper, and where the likelihood $p\left(\mathbf{Y}_{t_{1:n}}|\mathbf{W}, \mathbf{X}_{t_{1:n}}\right)$ is parameterised by a transformer $T_\mathbf{W}$ with weights $\mathbf{W}$. Plugging these terms in,

we verify that the BNAM does not produce consistent joint predictive distributions

$$\int p\left(\mathbf{Y}_{t_{1:n}}|\mathcal{D}_c,\mathbf{X}_{t_{1:n}}\right)\mathrm{d}\mathbf{Y}_{t_{m+1:n}} = \int\int p\left(\mathbf{Y}_{t_{1:n}}|T_{\mathbf{W}}(\mathbf{X}_{t_{1:n}})\right)q\left(\mathbf{W}|\mathcal{D}_c\right)\mathrm{d}\mathbf{W}\mathrm{d}\mathbf{Y}_{t_{m+1:n}} \quad (56)$$

$$= \int p\left(\mathbf{Y}_{t_{1:m}}|T_{\mathbf{W}}(\mathbf{X}_{t_{1:n}})\right)q\left(\mathbf{W}|\mathcal{D}_c\right)\mathrm{d}\mathbf{W} \quad (57)$$

$$= p\left(\mathbf{Y}_{t_{1:m}}|\mathcal{D}_c,\mathbf{X}_{t_{1:n}}\right) \quad (58)$$

$$\neq p\left(\mathbf{Y}_{t_{1:m}}|\mathcal{D}_c,\mathbf{X}_{t_{1:m}}\right) \quad (59)$$

and visualise the consequences of this in Fig. 10. By contrast, the BNNP models target outputs as conditionally independent given a weights sample for the MLP $f_{\mathbf{W}}$. This means the BNNP's likelihood decomposes as $p\left(\mathbf{Y}_{t_{1:n}}|f_{\mathbf{W}}(\mathbf{X}_{t_{1:n}})\right) = \prod_{i=1}^{n} p\left(\mathbf{y}_{t_i}|f_{\mathbf{W}}(\mathbf{x}_{t_i})\right)$ which in turn ensures consistency of the model through the fact that $p\left(\mathbf{Y}_{t_{1:m}}|\mathcal{D}_c,\mathbf{X}_{t_{1:n}}\right) = p\left(\mathbf{Y}_{t_{1:m}}|\mathcal{D}_c,\mathbf{X}_{t_{1:m}}\right)$. By the Kolmogorov extension theorem (Tao, 2011), consistency of a collection of joint distributions is needed for them to define a valid stochastic process, and it is for this reason that the BNAM does *not* define a stochastic process. Note that the other condition required is exchangeability; both the BNNP and BNAM exhibit this through permutation invariance with respect to the context observations and permutation equivariance with respect to the target inputs. While the lack of consistency is generally a disadvantage, in settings for which the target inputs are always the same, this behaviour of the BNAM would not matter. An example of such a scenario is the common NP task of image completion via pixelwise meta-regression—in this case the target inputs are always the complete set of pixel coordinates.

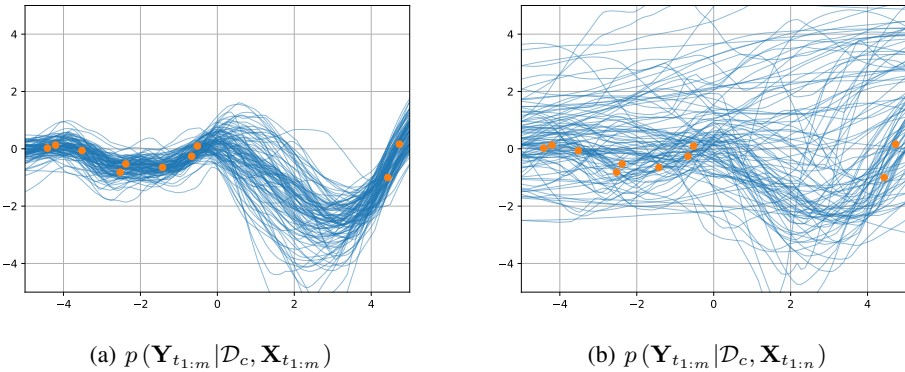

(a) $p\left(\mathbf{Y}_{t_{1:m}}|\mathcal{D}_c,\mathbf{X}_{t_{1:m}}\right)$          (b) $p\left(\mathbf{Y}_{t_{1:m}}|\mathcal{D}_c,\mathbf{X}_{t_{1:n}}\right)$

Figure 10: Inconsistent predictive distributions of a BNAM trained on GP-prior generated data. The orange dots are context observations and the wiggly blue lines are predictive samples. $\mathbf{X}_{t_{1:m}}$ is generated as a uniform grid of 100 locations between $-5$ and $5$ via `torch.linspace(-5.0, 5.0, 100)`, while $\mathbf{X}_{t_{1:n}}$ is generated by appending a value of 100.0 to $\mathbf{X}_{t_{1:m}}$ (i.e., $m = 100$ and $n = 101$). Observe that the joint distributions over $\mathbf{Y}_{t_{1:m}}$ are clearly very different. In other words, querying just a single extra target location can drastically change the BNAM's predictions, especially if the additional target location is OOD.

# F    EXPERIMENTAL DETAILS

The codebase can be found at github.com/Sheev13/meta-bdl.

## F.1    HOW GOOD IS THE BNNP'S APPROXIMATE POSTERIOR?

The BNN used to both generate and model the dataset had two hidden layers each with 20 units (i.e. overall architecture of [1, 20, 20, 1]) and ReLU nonlinearities. The prior was a zero-centered fully-factorised Gaussian with variances equal to the reciprocal of the number of input units for each layer. As described in the main text, the dataset was generated by sampling a set of weights from the BNN prior, uniformly sampling $N$ inputs $\mathbf{X}$ from $\mathcal{U}(-4.0, 4.0)$ where $N$ itself was sampled from $\mathcal{U}(21, 42)$, passing these inputs through the MLP parameterised with the sampled weights to obtain 24 corresponding clean outputs, and finally adding Gaussian noise of standard deviation 0.1 to each output to obtain the outputs $\mathbf{Y}$. The dataset is visualised in Fig. 11.

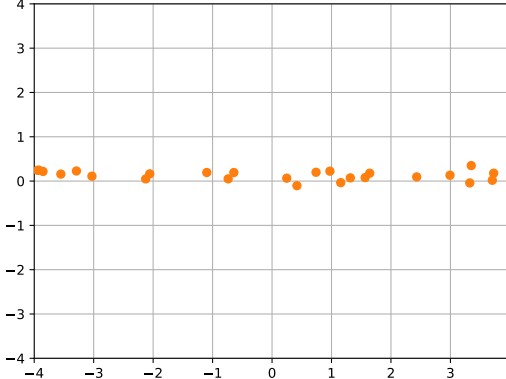

Figure 11: The synthetic dataset used in the investigation of approximate posterior quality.

For all approximate inference baselines, the standard deviation of the Gaussian likelihood was fixed to a particular value along the x-axes of Fig. 2, i.e. from `torch.logspace(-2, 1, 40)`, and the approximate inference algorithm was run four times using random seeds $[21, 42, 69, 420]$ and the below training parameters. The dataset was the same throughout.

**Meta BNNP.**    Two BNNPs were trained in this experiment; one as a meta-learner and one as a traditional learner. Both BNNP's had fixed $\Psi$, i.e. their priors were fixed to that of the data-generating BNN, and both used inference networks with two hidden layers of 50 units. The meta BNNP was trained using a meta-dataset of $50,000$ synthetically-generated datasets with identical generation process to the test dataset, save that the number of inputs was sampled from $\mathcal{U}(5, 50)$. The PP-AVI objective function was used, with the proportion of context points sampled from $\mathcal{U}(0.7, 0.9)$ with the remaining points being used as targets. A (meta-level) batch-size of 5 was used, and 8 samples from the posterior were used in each step for Monte Carlo estimates of the objective function. The optimiser was Adam under PyTorch defaults but with a learning rate that was linearly tempered from 5e-3 to 5e-5. The model was trained for $20,000$ steps, or equivalently (for a batch-size of 5 and $50,000$ datasets) 2 epochs.

**Non-meta BNNP and everything else.**    These models were trained by optimising the standard ELBO objective function that was estimated at each step using 8 samples from the posterior. All models were optimised via Adam under the PyTorch defaults and for $20,000$ steps of (within-task) full-batch training, i.e. for $20,000$ epochs. MFVI, UCVI, and LCVI used learning rates linearly tempered from 5e-3 down to 5e-5, FCVI from 5e-4 down to 5e-5, and GIVI from 1e-2 down to 5e-4 (it was much more stable to train, especially compared to FCVI). GIVI used 10 inducing points, with the inducing inputs initialised to a random (size-10) subset of the training inputs

**Evaluation.**    The obtained ELBO for each method was computed by estimating the ELBO with $10,000$ samples from the posterior. The naive Monte Carlo estimator of the log marginal likelihood used 10M samples (and the log-sum-exp trick). The KL divergences between approximate and true

posteriors were then computed by taking the difference between the log marginal likelihood and ELBO.

## F.2 DO MEANINGFUL BNN PRIORS EXIST?

### F.2.1 1D REGRESSIONS

For all data-generating processes in Fig. 3, BNNPs with two hidden layers of 64 units were used (i.e. overall architecture of $[1, 64, 64, 1]$), with inference networks also with two hidden layers of 64 units. In all cases, the Gaussian likelihood had initial standard deviation of $\sigma_y = 0.05$ but this parameter was jointly optimised with all inference network weights $\Theta$ and prior parameters $\Psi$. Throughout, a meta-level batch size of 5 was used.

**Sawtooth, Heaviside, standard BNN prior functions.** For these data-generating processes, $100,000$ synthetic datasets were generated where in each case the number of datapoints was sampled from $\mathcal{U}(40, 100)$ and the proportion of these points which were used as the context set was randomly sampled from $\mathcal{U}(0.1, 0.5)$. The PP-AVI objective function was optimised via Adam under PyTorch defaults, with a learning rate linearly tempered down from 3e-3 to 1e-4 for $50,000$ steps (making for 2.5 epochs of the meta-dataset), and where 32 posterior samples were used to estimate the objective function at each step. For the sawtooth data and standard BNN prior data, ReLU activations were used throughout. For the Heaviside data, tanh activations were used. Although the results are probably reproducible with any activation functions under a large enough architecture, we found in our finite-network-width setting that picking the "right" activation function for the problem was quite helpful. For the Heaviside setting, multiplying the posterior predictive and KL terms within the PP-AVI objective by $0.1$ yielded slightly nicer-looking prior-predictive samples. We hypothesise that this is because the true data-generating process does not sit within the set of those that can be represented by a BNN prior, perhaps due to the discontinuities when the function alternates between $-1.0$ and $+1.0$. Note that an infinitely wide BNN might be able to represent such stochastic processes in theory, but in our finite-width case this objective function tempering seemed to help.

The data generated from a standard BNN prior followed an identical generation procedure to that of Fig. 11, save that here the BNN architecture had hidden layer width of 64. The sawtooth data were obtained by randomly sampling inputs from $\mathcal{U}(-2.0, 2.0)$, obtaining a random sawtooth function via

$$f_{\text{saw}}(\cdot) = \varepsilon_1 \times \left( \frac{\cdot}{3} \right) + 0.25\varepsilon_2 + 1.33 \times \left( \text{remainder}(\cdot, 0.75) - \frac{0.75}{2} \right) \qquad (60)$$

where $\varepsilon_1$ and $\varepsilon_2$ were independently sampled from $\mathcal{N}(0, 1)$, and then adding Gaussian noise of standard deviation $0.05$ to the outputs of the random function. The Heaviside data were generated by sampling inputs from $\mathcal{U}(-5.0, 5.0)$, sampling a function from a GP prior with squared-exponential covariance function of unit lengthscale (and outputscale), passing the inputs through this function, subtracting the mean of the outputs from each of them, and then mapping all positive outputs to $+1.0$ and all negative outputs to $-1.0$. Finally, Gaussian noise of standard deviation $0.01$ was added to each output.

**ECG data.** The ECG data were synthetically generated using `neurokit2`'s `ecg_simulate` function. At a high level, for each dataset a single "heartbeat" was isolated via signal cropping and then placed at a random location within a six second window, with the resting voltage level used to pad on either side of the heartbeat to fill the window. More specific details are best conveyed via the following code snippet.

```
def generate_synthetic_ecg_task(
    fs=100,
    noise=0.001,
    n_range=None
):
    """Generate one synthetic ECG waveform and return (X, Y) torch tensors."""
    d = 6.0
    signal = nk.ecg_simulate(
        duration=8.0,
        sampling_rate=fs,
        heart_rate=20,
        noise=noise,
        method='simple'
    )
```

```
signal -= np.mean(signal)
signal /= (2*np.max(signal))

single_wave_block = signal[int(3.5*fs):int(6.5*fs)] # 3s long
zeros_block = np.concatenate(3*[signal[int(7.0*fs):]], axis=0) # 3s long
split = int(np.random.uniform(0.0, 3.0) * fs)
full = np.concatenate((zeros_block[:split], single_wave_block, zeros_block[split:]), axis=0)

# normalize and convert to tensors
t = np.linspace(-d/2, d/2, len(full), endpoint=False)
X = torch.tensor(t, dtype=torch.float64).unsqueeze(1)
Y = torch.tensor(full, dtype=torch.float64).unsqueeze(1)

if n_range is not None:
    n = torch.randint(low=min(n_range), high=max(n_range), size=(1,))
    inds = torch.randperm(len(full))[:n]
    return X[inds], Y[inds]

return X, Y
```

$100,000$ datasets were generated in this way for training. The PP-AVI objective function was estimated via 24 posterior samples at each step and it was optimised via Adam under PyTorch defaults with a learning rate linearly tempered from 1e-3 to 5e-5. Between 2 and 64 context points (uniformly chosen) were randomly selected from the total of $6 \times 100 = 600$ datapoints in each task, with the rest being used as targets. $500,000$ training steps were carried out, making for 25 epochs. This BNNP used tanh nonlinearities.

### F.2.2 IMAGE COMPLETIONS

The MNIST dataset of $28 \times 28$ pixel images of handwritten digits (LeCun et al., 1989) was used for this experiment. 2D regression datasets were generated by creating a 2D grid of coordinates $\mathbf{X}$ in $[-1.0, 1.0]^2$ and using each image pixel as a $y$ value corresponding to its respective grid coordinate $\mathbf{x}$. The $y$ values were mapped to the $[0, 1]$ range. Repeating this procedure for every image in the default MNIST training set resulted in a meta-dataset of $60,000$ with which to train a BNNP.

The BNNP's primary architecture consisted of four hidden layers of $64$ units, so the full sequence of layerwise activations was $[2, 64, 64, 64, 64, 1]$. The inference networks also had four hidden layers of $64$ units. ReLU nonlinearities were used throughout. Although the task was pixelwise regression, a Bernoulli likelihood was used since the outputs took values exclusively in $[0, 1]$. Note that this does not turn the problem into a classication problem since the output values were not strictly $0$ *or* $1$, but rather anywhere in the interval between $0$ and $1$ (inclusive).

The PP-AVI objective function was used to train the model, with the proportion of points used as context points uniformly chosen from the interval $[0.01, 0.6]$. Training was conducted over two episodes, each of $250,000$ steps and taking roughly 20 hours on an NVIDIA H100 GPU. The first episode saw the learning rate linearly tempered from 5e-4 down to 1e-5, and the second from 5e-5 down to 1e-6. Both episodes used 16 Monte Carlo samples, a meta-level batch size of 5. The purpose of the two training episodes was due to cluster usage rules limiting GPU job length to 24 hours.

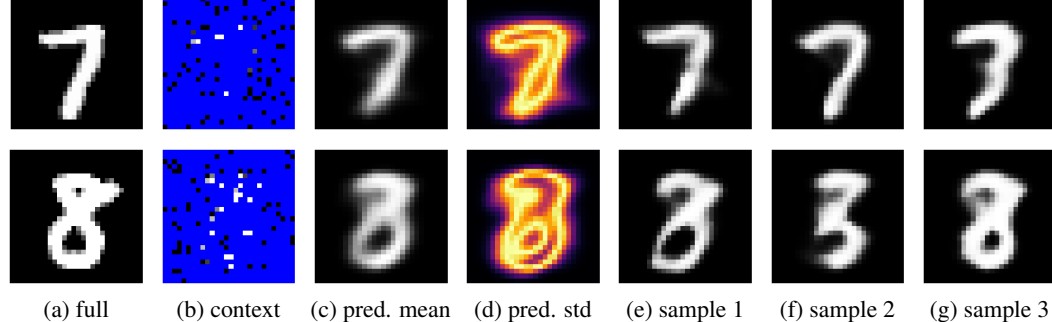

| (a) full | (b) context | (c) pred. mean | (d) pred. std | (e) sample 1 | (f) sample 2 | (g) sample 3 |

Figure 12: Two example image completion tasks executed by the trained BNNP. We see that with limited context data, the BNNP maintains a distribution over different types of digit that might explain the data; in both cases it is possible that the complete image could depict a three.

### F.3 DOES A GOOD PRIOR DOUBLE AS A LUNCH VOUCHER?

#### F.3.1 SYNTHETIC DATA SETTINGS

**Data.** The GP, Heaviside, and sawtooth training datasets were generated by uniformly sampling a number of datapoints from $\mathcal{U}(40, 100)$ and the sampled number of inputs were then uniformly sampled from $[-5.0, 5.0]$, $[-5.0, 5.0]$, or $[-2.0, 2.0]$ for the cases of GP, Heaviside, and sawtooth data respectively. The proportion of these points used as context points was sampled from $\mathcal{U}(0.1, 0.6)$ in all cases. For sawtooth and Heaviside data, the noiseless outputs were obtained in the same way as in Section F.2.1, and for GP data the noiseless outputs were obtained by sampling a function from a zero-mean and squared-exponential kernel GP prior with a lengthscale of $0.5$ and passing the inputs through the sampled function. The Heaviside data were corrupted by adding independently sampled Gaussian noise of standard deviation $0.01$ to the outputs, whereas the GP and sawtooth data were corrupted by Gaussian noise of standard deviation $0.05$. For the experiment itself, 16 datasets of each kind were generated once and saved, but with the difference that in all cases the context proportion was uniformly sampled from $\mathcal{U}(0.05, 0.5)$ and, where the input domain was previously $[-5.0, 5.0]$, it was changed to $[-4.0, 4.0]$.

**BNNP configuration and training.** The BNNP had two hidden layers of $48$ units in the primary architecture as well as the inference networks. We used SiLU, tanh, and ReLU activations for each of GP, Heaviside, and sawtooth data respectively. In each case, a meta-dataset of size $100,000$ was constructed and $100,000$ training steps performed over a meta-level batch size of $5$, giving a total of $5$ epochs. The PP-AVI objective function was used with 32 Monte Carlo samples, a learning rate that was linearly scheduled down from 3e-3 to 5e-5, and Adam under PyTorch defaults. The likelihood was a Gaussian whose variance was learned jointly with $\Psi$ and $\Theta$. Training took one or two hours on an NVIDIA A100 GPU.

**Baselines.** For each of the 16 test datasets, each approximate inference algorithm was run twice; once for a BNN with a standard prior (zero-mean, standard deviation equal to the inverse of the square root of layer width) and once for a BNN with the corresponding learned prior from the BNNP. All BNNs had the same architecture of two hidden layers of $48$ units and with the same nonlinearity as the BNNP in each data setting. Each approximate inference algorithm used a Gaussian likelihood with standard deviation $0.05$ and Adam under PyTorch defaults was used in all cases where optimisation was required. The remaining settings were as follows:

- *SWAG.* The network parameters were initialised by sampling from the prior before executing $5,000$ training steps with a learning rate of 5e-3. The SWAG algorithm itself (Maddox et al., 2019) was then used to compute an approximate posterior by updating the SWA statistics 100 times at every 25-th iteration of PyTorch default gradient descent (not Adam) with a learning rate of 1e-2. The rank of the low-rank correction to the diagonal covariance matrix, $K$, was $64$. The objective function was the unnormalised log posterior density of the training density.

- *MFVI, GIVI, and BNNP.* To ensure a fair comparison to the non-meta-models, the pre-trained BNNP itself was not used as a baseline; instead a new BNNP (with fixed likelihood noise of $0.05$) was trained in each case with the prior parameters $\Psi$ fixed and only the inference network parameters $\Theta$ set to trainable. For all three of these models, the (standard VI) ELBO was optimised for $75,000$ steps under a learning rate linearly tempered from 5e-3 down to 1e-4. $8$ Monte Carlo samples were used in estimating the ELBO via the reparameterisation trick. GIVI used $24$ inducing points, the inputs of which were initialised at a random subset of the context observations.

- *MCMC.* Because it allowed for easier tuning of hyperparameters, LMC was implemented as HMC with a single leapfrog integrator step for each proposal (rather than the usual discretised Langevin dynamics). HMC was performed using a step size of 5e-4 for $5,000$ steps each with 100 leapfrog integrator steps. We used a burn-in period of $500$ and thinned the samples by keeping only every 50-th sample. These hyperparameters were selected by looking at unnormalised log-posterior/log-likelihood/log-prior vs iteration plots and sample autocorrelation plots respectively. In the case of the learned prior for Heaviside data, a step size of 1e-4 was used instead. For LMC we used a step size of 1e-3 (except for learned

Heaviside prior, which had a stepsize of 5e-4) for $250,000$ steps. The burn-in and thinning parameters were $10,000$ and $5,000$ respectively, chosen via the same method as for HMC. A Metropolis-Hastings accept-reject step was performed after each new proposal, and in all cases the average acceptance frequency never fell below $0.99$ (which is how the step sizes were tuned). There was no minibatching.

### F.3.2 ERA5 SETTING

**Data.** The precipitation data were obtained from the ERA5 Land dataset (Muñoz Sabater et al., 2021). We focussed on an area of central Europe defined by coordinates $[50°, 5°, 45°, 12°]$ corresponding to the rectangle borders in the order of [N, W, S, E]. We used ten years of data from the beginning of 2010 through to the end of 2019. The grid granularities were $0.1°$ and 12 hours in the spatial and temporal axes respectively. The weather prediction task was spatial interpolation, where the goal is to impute missing grid values. The features used were 2D spatial coordinates and air temperature at 2m above ground-level, and the target was total precipitation in mm. All features and the targets were normalised, meaning each feature (and the targets) that were fed to the model had zero mean and unit standard deviation. Each timestep corresponded to a single task, giving a collection of about $\frac{24}{12} \times 365 \times 10 = 7300$ tasks, each with $\frac{50-45}{0.1} \times \frac{12-5}{0.1} = 3500$ total observations. Each task is therefore an image completion, save that the auxiliary information of ground air temperature was also provided to the model. A randomly chosen 16 datasets were reserved for testing, with the rest being used for meta-training the BNNP. For the training tasks, each time the task was encountered by the model (i.e. in different epochs) a random 25% of the observations were selected, and of the selected observations the context points were randomly selected in a proportion sampled from $\mathcal{U}(0.01, 0.5)$ with the rest serving as targets, meaning that during training each task had between $0.01 \times 0.25 \times 3500 \approx 9$ and $0.5 \times 0.25 \times 3500 \approx 438$ context observations. The subsampling was done purely for faster training. For the test tasks, the proportion of context points for each task was sampled from $\mathcal{U}(0.025, 0.25)$ and these points were selected randomly out of the points *not* in Switzerland. These splits were performed once and saved for the later evaluation across different models. The omitting of Swiss context points was done to ensure presence of a larger region without any context, in which the choice of prior would heavily influence predictions.

**BNNP configuration and training.** The BNNP had three hidden layers of $64$ units in the primary architecture as well as the inference networks, with ReLU activations throughout. The prior was learned by training the BNNP on the metadataset for $250,000$ training steps with a meta-level batch size of $5$, giving a total of roughly 70 epochs. The PP-AVI objective function was used with 24 Monte Carlo samples, a learning rate that was linearly scheduled down from 1e-3 to 1e-5, and Adam under PyTorch defaults. The likelihood was a Gaussian whose variance was learned jointly with $\Psi$ and $\Theta$. Training took around 22 hours on an NVIDIA H100 GPU.

**Baselines.** For each of the 16 test datasets, each approximate inference algorithm was run twice; once for a BNN with a standard prior (zero-mean, standard deviation equal to the inverse of the square root of layer width) and once for a BNN with the corresponding learned prior from the BNNP. All BNNs had the same architecture of three hidden layers of $64$ units and ReLU nonlinearities. Each approximate inference algorithm used a Gaussian likelihood with standard deviation $0.05$ and Adam under PyTorch defaults was used in all cases where optimisation was required. The remaining settings were as follows:

- *SWAG.* The network parameters were initialised by sampling from the prior before executing $10,000$ training steps with a learning rate of 1e-3. The SWAG algorithm itself (Maddox et al., 2019) was then used to compute an approximate posterior by updating the SWA statistics 200 times at every 25-th iteration of PyTorch default gradient descent (not Adam) with a learning rate of 5e-3. The rank of the low-rank correction to the diagonal covariance matrix, $K$, was $64$. The objective function was the unnormalised log posterior density of the training density.

- *MFVI, GIVI, and BNNP.* The treatment of BNNPs was the same as in the synthetic setting. For all three of these models, the (standard VI) ELBO was optimised for $75,000$ steps under a learning rate linearly tempered from 5e-3 down to 1e-4. 8 Monte Carlo samples were used in estimating the ELBO via the reparameterisation trick. GIVI used $24$ inducing points, the inputs of which were initialised at a random subset of the context observations.

- **MCMC.** For the same reason as in the synthetic setting, SGLD was implemented as SGHMC with a single leapfrog integrator step for each proposal (rather than the usual discretised Langevin dynamics). SGHMC was performed using a step size of 1e-4 for $5,000$ steps each with 100 leapfrog integrator steps. We used a burn-in period of $2,000$ and thinned the samples by keeping only every 50-th sample. These hyperparameters were selected by looking at unnormalised log-posterior/log-likelihood/log-prior vs iteration plots and sample autocorrelation plots respectively. For SGLD we used a step size of 5e-4 for $200,000$ steps. The burn-in and thinning parameters were $50,000$ and $2,500$ respectively, chosen via the same method as for SGHMC. No Metropolis-Hastings accept-reject step was performed, and a minibatch size of 128 was used in both cases. In cases for which the total number of context points was less than or equal to the batch size, the full batch was used.

### F.3.3 EVALUATION

In all data settings, the metrics of mean absolute error and log posterior predictive density were used. In both cases, they were computed on the target set in each of the 16 datasets after training. For computing the metrics, 1000 posterior samples were obtained from the variational and SWAG posteriors, and all the available samples after burn-in and thinning for the MCMC methods were used (60 samples in the ERA5 setting, 48 LMC samples and 90 HMC samples in the synthetic setting). The mean absolute error for each test task was computed as the absolute error between the true targets and the predictive mean (of the predictive samples), averaged over all target points in the task. These 16 scores were then averaged to give the final metric, with the standard error of these 16 values used as the errorbars. The log posterior predictive density for each task was computed as the likelihood density of the full target set averaged over posterior samples (i.e. Monte Carlo estimate of the posterior predictive integral), and then divided by the number of target points (for easier comparison between tasks with different number of targets). Of course, these computations were performed in the log-domain via the LogSumExp trick. These 16 scores were then averaged to compute the final metric for each method, with the standard error of the 16 values used as the errorbars. Note that for the MAE metric in the ERA5 setting, the data were transformed from the normalised domain to the original domain.

### F.3.4 CLOUDY WITH A CHANCE OF MEANINGFUL PRIORS

In this section we qualitatively compare precipitation patterns sampled from each of 1.) the real world (i.e., ERA5), 2.) a standard BNN prior, and 3.) our BNNP-learned prior. The comparison is shown in Fig. 13. We see from the examples in the left-hand column that real-world precipitation can be somewhat spatially sparse, with much of the area receiving no precipitation. This behaviour is reflected in the samples from the learned prior in the right-hand column, but *not* in those of the standard BNN prior. Furthermore, precipitation values must be non-negative—this is observed in the real-world examples as well as those of the learned prior, but not for the standard prior (see the minimum values on the colour-value key for each example). Finally, the alps cause variations in temperature with a relatively high spatial frequency (on the scale of our area), and the models are given realistic settings of this feature from the ERA5 dataset. It is the variation of this feature that causes the rough patterns over southern Switzerland in the sampled precipitation patterns from the standard BNN prior, which appears to be overly/naïvely sensitive to it. By contrast, the BNNP seems to have learned a prior that models this feature more appropriately, most likely by encoding the interaction between this feature and the location. For example, we know that altitude varies across our region and that precipitation depends on both temperature *and* altitude, suggesting the temperature feature's importance might vary across the region. We suspect the BNNP's learned prior incorporates such a notion of spatially-variable feature importance.

It is clear that the learned prior more faithfully models real-world precipitation behaviour than the standard BNN prior does. Indeed, a model for precipitation that allows for negative values—raindrops flying up from the earth and into the clouds—is somewhat questionable. It is no wonder, then, that predictive performance is also superior when using the learned prior than when using a naïve one (Fig. 6). In general, we would encourage researchers and practitioners to question their use of standard BNN priors, and to consider the possibility of meta-learning their priors instead.

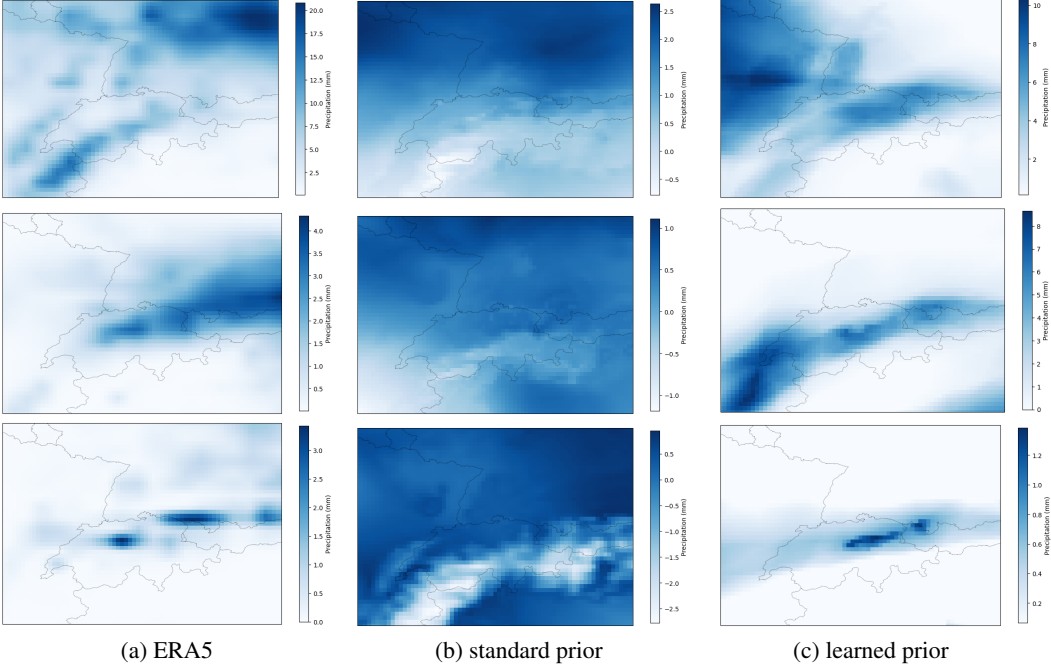

|  (a) ERA5 | (b) standard prior | (c) learned prior |

Figure 13: Sampled precipitation patterns over an area of central Europe centered on Switzerland. The samples from the learned BNN prior are much more realistic than those of the standard BNN prior.

### F.4 CAN A RESTRICTED PRIOR IMPROVE META-LEVEL DATA EFFICIENCY?

### F.4.1 DATA

**Abalone.** We downloaded the Abalone dataset (Nash et al., 1994) from the UCI repository. These data have 8 input features such as height, diameter, shucked weight. One of the features is the specimen's sex, which is one of three values: male, female, or infant. We separated the data into three

partitions based on the value of the sex feature, and then discarded this feature in all cases to leave only 7 features. All input features were then normalised to have zero mean and unit standard deviation, and this was done globally (rather than within each of the three datasets). The output variable for each specimen is the number of rings, which is closely related to the age of the specimen. This variable was globally normalised in the same way as the features. The datasets corresponding to male and female specimens were used as the meta-dataset for training the meta-learners, so $|\Xi| = 2$. The male and female datasets had 1528 and 1307 observations respectively. The dataset corresponding to infant abalones was used as the test dataset, and it contained 1342 points. Similarly to how the ERA5 test data included a large region (Switzerland) with no context points in order to emphasise the impact of the prior on predictions, in the infant abalone daaset we constructed a hypercube in the feature space that contained roughly $20\%$ of points and set all of these to targets. Of the remaining points, a random selection were selected as context points such that the overall proportion of context points was 0.25. This split was performed once and saved so that all baselines were evaluated on the test task with the same context/target split.

**Paul15.** We downloaded the Paul15 mouse bone marrow cell dataset (Paul et al., 2015) using `scanpy` (Wolf et al., 2018). The raw dataset contains roughly $3,400$ features, where each one represents the degree to which a certain gene is expressed. After normalisation, we used PCA to map this down to the top 100 eigen-genes. The target feature for this dataset is the *pseudotime*, a value computed by biologists that serves as a metric for cell age/development stage. A young stem cell will have a low pseudotime, whereas a cell that has already specialised for a certain role will have a higher pseudotime. We normalise the psuedotime feature as well. Paul et al. (2015) provide a partitioning of the data into 19 clusters, broadly corresponding to cells of the same type/specialisation (or lack thereof for young cells). We use these clusters to convert the problem into a meta-learning one. We randomly selected 10 of the clusters to use as the training meta-dataset, and of the remaining 9 we selected the task with the most datapoints to use as the test task. This test task was split into context and target sets with a context proportion of 0.2, and this split was saved and used for all baselines. The numbers of datapoints in the 10 training tasks were 153, 69, 373, 164, 9, 43, 246, 180, 167, and 63. The test task had a total of 329 datapoints.

### F.4.2 MODELS

**BNNP.** For the Abalone setting the BNNP had three hidden layers of 64 units, as did its inference networks. For the Paul15 setting, there were four hidden layers of 96 units throughout instead. SiLU nonlinearities were used in both cases. The likelihood was Gaussian with a learned standard deviation that was initialised to 0.1. The *prior-learnability* decimal represents the proportion of all the parameters of the prior $\Psi$ for which `parameter.requires_grad` was set to `True`. Each weight in the BNNP's primary architecture has a prior mean, a prior variance, and prior covariances with the other weights corresponding to the same output neuron. To set a fixed proportion $p$ of these to trainable, we found the total number of weights $|\mathbf{W}|$, found the nearest integer to $p|\mathbf{W}|$, and then, proceeding through all weights in the network from the first layer through to the last, we set the prior parameters associated with each weight to trainable until we reach the integer. Training was performed by optimising the PP-AVI objective. The objective was estimated from 16 Monte Carlo samples at each step, and optimised for $100,000$ steps. For the Abalone setting, the meta-level batch size was 2 (i.e. full-batch training) but for Paul15 it was 5, meaning training was performed for $100,000$ epochs for the Abalone setting and $50,000$ for the Paul15 one. In both cases, every time a training dataset was encountered, it was randomly split into a new context-target partition where the proportion of context points was itself randomly sampled from $\mathcal{U}(0.1, 0.6)$ each time. The optimiser was Adam under PyTorch defaults, with a learning rate that was linearly scheduled down from 5e-4 (Abalone) and 1e-4 (Paul15) to 5e-5.

**Baselines.** In all cases, optimisation was performed with Adam under PyTorch defaults.

- *BNNs.* As with the BNNP, SiLU nonlinearities were used in all cases, and the BNNs had three hidden layers of 64 units in the case of the Abalone data, but four hidden layers of 96 units in the case of the Paul15 data. GIVI had 128 inducing points in the Abalone setting and 196 in the Paul15 one, and in both cases the inducing inputs were initialised to a random subset of the test set context set. As non meta-learners, these models were trained only on the test-set context set in each case. As demonstrated in Bui (2021) or in our first experiment,

GIVI's approximate posterior is of high enough quality that the ELBO can be used for model selection, and so the Gaussian likelihood's standard deviation was learned in the case of GIVI (initialised to $0.1$), and the learned/optimal values were then used for MFVI (but frozen in place). The standard variational ELBO was optimised for $75,000$ steps, each time being estimated from $8$ Monte Carlo samples. The learning rate was linearly tempered down from 5e-3 to 1e-4.

- **NP and BNP.** These models used encoders and decoders each with three hidden layers of $256$ units and SiLU nonlinearities throughout. They used a Gaussian likelihood with a standard deviation initialised to $0.1$ but optimised with the rest of the parameters. The objective function was the average target-set log posterior predictive density (see Section A.3), which, for each task, was estimated from $16$ latent variable posterior samples. The meta-level batch sizes were $2$ for the Abalone setting and $5$ for the Paul15 one. Optimisation was performed for $500,000$ iterations with a learning rate linearly tempered down from 1e-4 to 1e-5. These models were very slow to converge, hence the very high number of training steps. Every time a training dataset was encountered, it was randomly split into a new context-target partition where the proportion of context points was itself randomly sampled from $\mathcal{U}(0.1, 0.6)$.

- **AR-TNP.** This model consisted of three self-attention blocks where every layer had dimensionality of $256$ and SiLU nonlinearities were used throughout. The objective function was again the average target-set log posterior predictive density, but as a type of conditional NP, the model outputs the marginals for each target and so no sampling was required, just multiplication of the (independent) marginals. The meta-level batch sizes were $2$ for the Abalone setting and $5$ for the Paul15 one. Optimisation was performed for $50,000$ steps (this model reached convergence much quicker than the other NPs), and every time a training dataset was encountered, it was randomly split into a new context-target partition where the proportion of context points was itself randomly sampled from $\mathcal{U}(0.1, 0.6)$. Not that this model was trained in the same way as a regular TNP(-D).

### F.4.3 EVALUATION

For all models, training was repeated four times for seeds $[21, 42, 69, 420]$. In each of the four cases and for all models, $1000$ predictive samples were used to compute the metrics, where for AR-TNP the samples were obtained via the autoregressive scheme developed by Bruinsma et al. (2023). For each data setting, the metrics were computed as follows. The mean absolute error for a test task was computed as the absolute error between the true targets and the predictive mean (of the predictive samples), averaged over all target points in the task. The 4 scores obtained across the four seeds were then averaged to give the final metric, with the standard error of these 4 values used as the errorbars. The log posterior predictive density for each task was computed as the likelihood density of the full target set averaged over posterior samples (i.e. Monte Carlo estimate of the posterior predictive integral), and then divided by the number of target points (for easier comparison between tasks with different number of targets). Of course, these computations were performed in the log-domain via the LogSumExp trick. These 4 scores were then averaged to compute the final metric for each method, with the standard error of the 4 values used as the errorbars. Note that for the MAE metric, the data were transformed from the normalised domain to the original domain.

