# OpenReview forum: "Amortising Inference and Meta-Learning Priors in Neural Networks"
_ICLR.cc/2026/Conference — ICLR 2026 Poster_

### Official Review · Reviewer_xjWL · 2025-10-22

**Soundness:** 3
**Presentation:** 3
**Contribution:** 3
**Rating:** 4
**Confidence:** 4

**Summary:**

The paper introduces a meta-learning model to infer better priors for Bayesian neural nets (BNNs) using a neural process (NP)-based approach, which treats the BNN's weights as latent variables.
To ensure scalability, it relies on layerwise factorizations and introduces within-task minibatching.

**Strengths:**

- The paper proposes an interesting and novel combination of BNNs and NPs for an important unsolved task in the Bayesian deep learning literature.
- The proposed objective (eq. 10) consists of interpretable and well-motivated terms.
- The approach is scalable and seems to provide promising results across a range of experiments.

**Weaknesses:**

- Proposition 1 is promising in that the objective is well-motivated but does not provide any guarantees that the vague definition of "small" in Definitions 1–3 is actually achievable. For example, are all three terms minimizable at the same time, or do they balance each other?
- The empirical evaluation is promising but currently rather unusable, as it is completely devoid of any information on architectures, hyperparameters, training procedures, etc.
- No code is provided.

## Minor weaknesses
- Hiding the definitions of the three desiderata (167–170) in the appendix makes that part of the paper feel rather handwavy, even though it isn't.
- Details are missing from the results, e.g., over how many seeds are the error bars in Figure 6?

**Questions:**

- Q1: The paper doesn't discuss computational cost and runtime requirements. How expensive is the approach?

---

> ### Author Response · Authors · 2025-11-28
> **Response to Reviewer xjWL's Review**
>
> Thank you for your balanced review. We find it encouraging that you find our combination of BNNs and NPs to be both interesting and novel, and we address your comments below.
>
> ### Weaknesses
>
> > Proposition 1 is promising in that the objective is well-motivated but does not provide any guarantees that the vague definition of "small" in Definitions 1–3 is actually achievable. For example, are all three terms minimizable at the same time, or do they balance each other?
>
> Since our proposition merely states that our objective function "targets" the desiderata rather than e.g. "achieves" them, we find it unnecessary to concretely define what we mean by "small". As we said in our response to a similar comment from reviewer `G27C`, we would be happy to change our definitions to relative ones if this would resolve the issue for you?
>
> We completely agree that whether or not a sufficiently large BDNP is capable of achieving "smallness" of these terms would certainly be of interest, and especially to those wishing to use the BDNP as a general-purpose probabilistic meta-learner. However, we do not present the BDNP as a tool for such a purpose and this work's focus is therefore not in studying the BDNP itself. As you can see in appendix A or in our response to reviewer Kn36's first question, we intuitively expect the two terms in our PP-AVI objective to share the same argmax. Theoretical confirmation that a large enough BDNP can achieve simultaneous "smallness" of these terms would be a promising result, and we hope to pursue this in future work.
>
>
> > The empirical evaluation is promising but currently rather unusable, as it is completely devoid of any information on architectures, hyperparameters, training procedures, etc.
>
> This is a highly valid critisicm, and we apologise for not including such experimental details in time for the submission deadline. We have now included all experimental details in a new and comprehensive appendix section which you can see in the revised version of the manuscript.
>
> > No code is provided.
>
> We intend to make our codebase public upon publication of this work, at which point we will also make the codebase easily accessible from the manuscript.
>
> ### Minor Weaknesses
>
> > Hiding the definitions of the three desiderata (167–170) in the appendix makes that part of the paper feel rather handwavy, even though it isn't.
>
> We completely agree that the partitioning of Proposition 1 in section 2.3 from the definitions needed to prove it in appendix A.1 results in a decreased sense of rigour for first-time readers. However, we do this plainly because of space constraints; our three definitions take up almost an entire page and there is simply not enough space in the main text to include them. Perhaps we can resolve this by mentioning in the main text that we provide concrete definitions of the three desiderata in the appendix---would this be sufficient to resolve your concern here?
>
> > Details are missing from the results, e.g., over how many seeds are the error bars in Figure 6?
>
> In Figure 6's experiment, we pretrained a BDNP once for each of the four data generating processes in order to obtain four corresponding well-posed BNN priors. For each data-generating process, we generated and saved 16 datasets, each with fixed context-target splits. For each of these 16 datasets and in each of the four data settings, each approximate inference algorithm was used to approximate the posterior of a BNN given only the context set and the metrics were computed on the target set. These metrics were then normalised by the number of target points and averaged across the 16 datasets. The standard deviation of the sample mean of the metric was used as the errorbar size in each case.
>
> Such details are now included in our new Experimental Details appendix too.
>
> ### Questions
>
> > Q1: The paper doesn't discuss computational cost and runtime requirements. How expensive is the approach?
>
> We apologise for any misunderstanding here. If we can draw your attention to the penultimate paragraph of Appendix C (Minibatching), you will see a discussion of the computational cost associated with the BDNP. The BDNP is scalable except for in the architecture width, and we point this out in Section 5 as a reason for why we do not present the BDNP as a practical and general-purpose NP.
>
> ### Conclusion
>
> We are most appreciative of the time and effort you have evidently devoted to reviewing our work, and we would be very grateful if you would consider increasing your score if our responses have adequately addressed your concerns. Otherwise, we look forward to engaging in further discussion with you to improve our paper even more.

---

### Official Review · Reviewer_poex · 2025-10-31

**Soundness:** 3
**Presentation:** 3
**Contribution:** 3
**Rating:** 6
**Confidence:** 4

**Summary:**

This paper addresses the challenge of defining useful priors in a Bayesian deep learning setting by introducing the
Bayesian Deep Neural Process (BDNP). The central idea is to meta-learn a parametric prior for Bayesian Neural Network
(BNN) weights, coupled with amortized variational inference, using a meta-dataset of related tasks. Conceptually, the
BDNP can be viewed as a neural process (NP) where the latent variable represents the weights of a BNN, and the decoder
is the BNN itself. The paper presents compelling empirical evidence demonstrating the BDNP's ability to achieve
high-quality approximate inference compared to other VI methods and to learn meaningful priors effectively.

**Strengths:**

The presentation of the paper (language and structure) is well done. The theoretical section is sound and nicely concise.
The experimental section is comprehensive and well-executed. Presented ablation studies, particularly those evaluating the
quality of approximate inference against other VI methods and the qualitative and quantitative analysis of learned priors,
strongly support for the paper's claims. The authors provide a commendably clear and realistic discussion of the
limitations of their approach, such as potential scalability issues for very large or complex architectures.

**Weaknesses:**

- **Training Loss Justification:** The paper's core training objective, PP-AVI, deviates from standard NP-losses like NP-VI.
While Appendix A.4 provides a detailed justification for this choice over NP-VI, this reasoning should be
(at least partly) integrated into the main text. Furthermore, the PP-AVI 'loss' is formulated as a maximization
objective rather than a loss.
- **Clarification on Within-Task Minibatching:** The claim that "the ability to minibatch a forward pass over a given
context set is rare in the context of neural processes" requires more elaboration. The authors acknowledge
that inference in BDNP can be viewed as Bayesian context aggregation in NPs [1]. Consequently, iteratively updating the
latent posterior over multiple mini batches via sequential Bayesian inference is also applicable to classical NPs.
This also works for other context aggregation mechanisms such as mean-aggregation or max-aggregation. Only context
aggregation in the Attentive NP can not be minibatched, but the same holds for tha Attentive BDNP (AttBDNP). Furthermore,
the method of using gradients from a random minibatch during training (Appendix C) is a significant detail. This
approximation introduces potential biases and its implications (e.g., impact on convergence, generalization) should be
more prominently discussed in the main text. Additionally, the assertion that "our scheme maintains the ability for the
BDNP to learn to generalize predictive inference across various context set sizes during training" requires theoretical
or empirical evidence to support it.
- **Evaluation Metrics for Correlated Samples:** The per-datapoint Log Posterior Predictive Density (LPPD) is used as one
primary metric without a formal definition in the paper. More importantly, a key advantage of latent NPs (including BDNPs)
over conditional NPs (CNPs) is their capacity to produce correlated function samples. LPPD, being a per-datapoint metric,
may not fully capture or adequately evaluate this crucial aspect of model calibration and uncertainty representation.
It is recommended to also evaluate the posterior predictive log-likelihood over the entire target set, i.e.,
$p(y^{1:M}_T\mid x^{1:M}_T, D_C)$, as this metric would provide a more holistic assessment of the model's ability to
generate coherent and well-calibrated correlated function samples.

[1] Volpp et al., 'Bayesian Context Aggregation for Neural Processes'

**Questions:**

- In the experiments comparing BDNPs to standard NPs, did you also learn the prior for the latent variable?

---

> ### Author Response · Authors · 2025-11-28
> **Response to Reviewer poex's Review. Part 1.**
>
> We thank you for your thoughtful review and are very appreciative of the constructive critisicm. We are encouraged that you find our experiments to be comprehensive and to strongly support the claims we make in the paper. We do our best to address your questions and concerns below.
>
> ### Weaknesses
>
> > Training Loss Justification
>
> Both you and reviewer `G27C` make this request, and we agree that developing the main-text discussion/introduction of the objective function would benefit the paper. We have updated the manuscript to reflect this, and we have uploaded our new manuscript in tandem with these rebuttals.
>
> Regarding our formulation of the loss as a maximisation objective---well spotted and many thanks for pointing this out. We have updated the manuscript to ensure consistency in this regard.
>
> > Clarification on Within-Task Minibatching
>
> You are correct that within-task minibatching is also possible in classical NPs such as the original CNP, latent-variable NP, or Volpp et al.'s scheme. However, it is worth pointing out that these models have not represented the state-of-the-art in probabilistic meta-learning for some time, with NPs such as Kim et al. (2019)'s ANP, Gordon et al. (2019)'s ConvCNP, or Nguyen and Grover (2022)'s TNP being of higher practical relevance due to their far superior performance. As with the ANP that you pointed out, ConvCNP-based models (e.g. ConvCNP, ConvGNP, ConvNP) and TNP-based models (e.g. TNP-D, TNP-ND, EQTNP, LBANP, TE-TNP) are *not* amenable to within-task minibatching. Since these NPs are much more relevant to practitioners than the relatively poor-performing classical NPs, we opted to summarise this in the paper by just describing within-task minibatching as "rare" in NPs, a statement which we continue to stand by. We would be happy to include some of this discussion in the paper itself if you believe readers would benefit from it.
>
> Regarding our method of using gradients from just one minibatch to enable *training* under our minibatching scheme, we completely agree that this approximation could introduce new biases as is typically the case with stochastic optimisation. Although we never needed to use this scheme in our experiments, we did conduct a few minibatched training runs out of curiosity and found that training still worked well but it benefitted from a slightly smaller learning rate. Since this outcome is typical of stochastic optimisation, we originally felt there was no need to highlight this limitation in the main text, especially since it is not applicable when using the minibatching scheme at prediction time. We do now refer to this matter in the main text, which you can see in our revised manuscript.
>
> Finally, you make a very good point that this assertion is perhaps too strong given the lack of evidence to support it. In the revised manuscript we have updated this statement to be more obviously speculative.
>
> > Evaluation Metrics for Correlated Samples
>
> This is also an excellent point. Indeed the per-dataset posterior predictive density will rightly reward models for which inter-target correlations are modelled well, whereas a per-datapoint one will not. We apologise for not including any experimental details in the original submission, but we have now added all experimental details in a new appendix in the revised version of the manuscript. In this section, you will see that the way we compute our "per-datapoint" log posterior predictive density is actually by computing the per-data**set** one and dividing by the number of target points, meaning the way this metric is implemented already reflects the benefits of predicting coherent function samples. We divided by the number of target points for ease of comparison across settings with different target set sizes.
>
> As an aside, note that all models used in the paper actually produce coherent function samples rather than marginal predictives. This is not by accident; we did this to ensure the baselines were as relevant as possible to the BDNP. For example, if a BDNP performed better than a TNP-D, this could be explained (at least in part) by the fact that the TNP-D cannot model correlations between predictions, meaning it would be harder for us to conclude that it is the learned prior that makes the difference.

---

> > ### Author Response · Authors · 2025-11-28
> > **Response to Reviewer poex's Review. Part 2.**
> >
> > ### Questions
> >
> > > In the experiments comparing BDNPs to standard NPs, did you also learn the prior for the latent variable?
> >
> > We did not. All NPs were trained by optimising the target-set posterior predictive (following what is proposed in [2] and also commonly done in modern latent-variable NPs such as [3]). Under such a training scheme (but also under the NPVI training objective) no explicit formulation of the prior is ever required in computing the objective. For the BNP, an explicit parameterisation is indeed required for computing the posterior over the latent variable, but this is *not* learned, it is fixed as a zero-mean, unit-variance, fully factorised Gaussian. However, note that it is never necessary to learn the prior in such an NP---the encoder in trained to map from the data space to a latent space in which representations are Gaussian distributed, and similarly the decoder is trained to map from this Gaussian-distributed latent space back to the data space. Learning the prior in such an NP would be a little bit like learning the prior in a VAE or learning the base noise distribution in a diffusion model---we would expect no benefit, assuming the architecture is sufficiently flexible.
> >
> > ### Conclusion
> >
> > Thanks once again for the time and energy that you have evidently devoted to this review of our work. We hope our comments have sufficiently addressed your concerns or eliminated any misunderstandings that we may have caused. We look forward to hearing back from you, and we invite you to notify us of any further concerns so that we can work towards improving the paper even further. We would be very grateful if you would consider increasing your score if you deem our responses and updated manuscript to be so deserving.
> >
> > [2]. Gordon, J., Bronskill, J., Nowozin, S., Bauer, M., & Turner, R. E. (2019, January). Meta-learning probabilistic inference for prediction. In 7th International Conference on Learning Representations, ICLR 2019.
> >
> > [3]. Foong, A., Bruinsma, W., Gordon, J., Dubois, Y., Requeima, J., & Turner, R. (2020). Meta-learning stationary stochastic process prediction with convolutional neural processes. Advances in Neural Information Processing Systems, 33, 8284-8295.

---

### Official Review · Reviewer_Kn36 · 2025-11-01

**Soundness:** 3
**Presentation:** 3
**Contribution:** 3
**Rating:** 4
**Confidence:** 3

**Summary:**

This work aims to incorporate a meta-learning prior into the weight parameters of Bayesian Neural Networks (BNNs). In doing so, it builds BNNs that can be interpreted as a type of latent neural process model, where the latent variable is replaced by the posterior distribution of the BNN’s weights, which varies depending on the input. Specifically, it introduces an inference network that generates pseudo-labels for each linear layer and updates the layer’s posterior using these pseudo-labels and the prior. Once the parameters of the inference network and the BNN prior distribution are learned, indicating that the BNN has a well-posed prior, it evaluates the corresponding BNN's ability to perform various tasks effectively. Furthermore, it investigates the crucial role of approximate Bayesian learning under a well-posed prior.

**Strengths:**

* This work introduces an interesting idea of incorporating a meta-learning prior into BNNs and establishes a connection to neural processes.
*  It also presents an amortized linear layer structure that enables each layer to function as a Bayesian layer with an amortized prior.

**Weaknesses:**

* Although the proposed method is technically sophisticated, it is unclear what specific problem it aims to address with the proposed structure. For instance, it is not evident whether the main contribution lies in emphasizing the amortized prior for BNNs and its benefits, or in investigating approximate training for BNNs with a well-chosen prior.

* The training procedure for the parameters of the inference network and the prior distribution is insufficiently described. Beyond presenting the loss objective, a pseudo-code illustration or detailed algorithmic explanation would be necessary for clarity.

* Moreover, the quality of the learned prior and the corresponding posterior is likely influenced by the design of the inference network. However, this aspect appears to be underexplored in this work.

**Questions:**

* It appears that the parameters of the inference network and the prior distribution are jointly trained according to Eq. (10). The ELBO term seems primarily used for training the parameters of the prior distribution, while the conditional likelihood for the target task appears to update both the inference network and the prior distribution. Is this interpretation correct? If not, could you please clarify this point further?

* In this training procedure, was the inference network pre-trained, or was it trained jointly from scratch?

* How is the scale of prior learnability (ranging from 0.0 to 1.0) measured in Figure 7? Moreover, if the amount of data is limited and the prior is well-posed, shouldn’t the prior be beneficial? Why does the performance of BDNP with a prior-learnability of 1.0 degrade so significantly in Figure 7(a)?

* The following works seem relevant to the concept of meta-priors for BNNs and neural processes, and might be useful to include in the related work section:

Hierarchical Gaussian Process Priors for Bayesian Neural Network Weights, NeurIPS 2020

Bayesian Convolutional Deep Sets with Task-Dependent Stationary Prior, AISTATS 2023

---

> ### Author Response · Authors · 2025-11-28
> **Response to Reviewer Kn36's Review. Part 1.**
>
> Thanks for your detailed feedback and for contributing to a constructive discussion regarding our work. We do our best to address your concerns below.
>
> ### Weaknesses
>
> > Although the proposed method is technically sophisticated, it is unclear what specific problem it aims to address with the proposed structure. For instance, it is not evident whether the main contribution lies in emphasizing the amortized prior for BNNs and its benefits, or in investigating approximate training for BNNs with a well-chosen prior.
>
> We view our contribution as one of primarily scientific value but with some engineering value too. The BDNP enables us to answer questions that are of interest to the Bayesian deep learning community and that were previously not possible to answer. For example, in our work we show that
> 1. well-specified priors *do* exist for BNNs (and it's possible to learn them),
> 2. Gaussian priors in BNNs can be highly flexible,
> 3. a good BNN prior does not mean our lunch is free; high-quality approximate inference is *still* necessary for BNNs to work well.
>
> These are important findings that meaningfully progress the state of knowledge in the Bayesian deep learning community. In addition, the BDNP provides a practical solution to the problem of poor generalisation in the context of data-limited meta-learning settings, as well as to the problem of scaling neural processes to large context sets. We are aware that our paper delivers both scientific and engineering contributions, but we would implore you to view this as a strength rather than a weakness.
>
> > The training procedure for the parameters of the inference network and the prior distribution is insufficiently described. Beyond presenting the loss objective, a pseudo-code illustration or detailed algorithmic explanation would be necessary for clarity.
>
> This is a fair criticism, and we are happy to add a block of pseudocode to the manuscript to further elucidate the training procedure. We outline our training algorithm as follows:
> - **Require**: meta dataset $\Xi$, meta-level batch size $M$, number of training steps $N$, number of Monte Carlo samples $k$
> - **For** training step up to $N$:
>     - randomly select batch of M tasks from $\Xi$
>     - batch_loss $\leftarrow 0.0$
>     - **For** each task $\mathcal{D}^{(j)}$ in batch of tasks:
>         - compute layerwise posteriors and obtain $k$ samples using algorithm 1 in appendix C
>         - compute a $k$-sample estimate of $\mathcal{L}_\text{PP-AVI}\bigl(\mathcal{D}^{(j)}\bigr)$ using the practical tips at the bottom of appendix A.1.
>         - batch_loss = batch_loss $- \frac{|\Xi|}{M}\mathcal{L}_\text{PP-AVI}\bigl(\mathcal{D}^{(j)}\bigr)$
>     - Conduct step of gradient descent on batch_loss w.r.t. $\Theta$ and $\Psi$ using an update rule of your choosing (e.g. Adam).
>
> This has been added in Appendix A.6.
>
> > Moreover, the quality of the learned prior and the corresponding posterior is likely influenced by the design of the inference network. However, this aspect appears to be underexplored in this work.
>
> It is true that we did not ablate the design of the inference networks in our study. The reason for this is that we prioritised research questions that were of broader scope than developing a better understanding of the BDNP itself; we focussed on what the BDNP enables us to learn about Bayesian neural nets and the unique ways in which it advances the neural process field. We agree that such an ablation study could be useful to practitioners who wish to use a BDNP, but ultimately this ablation joins a fairly long list of exciting research opportunities that we hope to tackle in future work:
> - inference network design ablation,
> - theoretical analysis of BDNP training under our within-task minibatching scheme,
> - benchmarking of BDNPs and BDAMs as general-purpose probabilistic meta-learners,
> - benchmarking of BDNPs as general-purpose probabilistic (meta) online learners using the scheme from appendix D,
> - theoretical analysis of large NNs with Gaussian weights being universal stochastic process approximators (i.e. whether a BNN prior with Gaussian weights can approximate any stochastic process in theory)
> - empirical ablation of the different degrees of Gaussian prior factorisation in appendix B
> - rigorous ablation of different objective functions on quality of approximate inference as well as quality of learned prior.

---

> > ### Author Response · Authors · 2025-11-28
> > **Response to Reviewer Kn36's Review. Part 2.**
> >
> > ### Questions
> >
> > > It appears that the parameters of the inference network and the prior distribution are jointly trained according to Eq. (10). The ELBO term seems primarily used for training the parameters of the prior distribution, while the conditional likelihood for the target task appears to update both the inference network and the prior distribution. Is this interpretation correct? If not, could you please clarify this point further?
> >
> > In principle, either term in isolation could be used to train both families of parameters. As reviewer `G27C` points out, optimising the ELBO term is similar in spirit to conducting a type-II maximum likelihood/empirical Bayes procedure. So, while of course encouraging the approximate posteriors (parameterised by $\Psi$) to approach the true ones, it simultaneously drives the prior (parameterised by $\Theta$) to the one for which the training datasets are "least surprising" (since $p(\mathbf{Y}_c|\mathbf{X}_c)$ is maximised on average over the tasks). Under a good prior and with accurate approximate posteriors, we would expect the posterior predictives to be of high quality as well.
> >
> > On the other hand, optimising the posterior predictives (parameterised by both $\Psi$ *and* $\Theta$) directly should naturally encourage high quality posterior predictives. This procedure has already been studied as an alternative to variational inference in the meta-learning setting [2], and the authors show that the procedure is sufficient for learning a well-specified implicit prior along with high quality posteriors. While this setting is slightly different to ours since our priors are explicit, we would expect the conclusion to hold in our setting.
> >
> > As discussed in appendices A.2 and A.3, although either term might be sufficient in theory, in practice we found this not to be the case. Refer to these appendices for our speculation on why we think this is the case, and why using both terms together seems to fix the problems caused by using just one term.
> >
> > > In this training procedure, was the inference network pre-trained, or was it trained jointly from scratch?
> >
> > The inference networks were trained jointly with the prior parameters, and from scratch.
> >
> > > How is the scale of prior learnability (ranging from 0.0 to 1.0) measured in Figure 7? Moreover, if the amount of data is limited and the prior is well-posed, shouldn’t the prior be beneficial? Why does the performance of BDNP with a prior-learnability of 1.0 degrade so significantly in Figure 7(a)?
> >
> > The decimal represents the proportion of all the parameters of the prior $\Psi$ for which `parameter.requires_grad` was set to `True`. Each weight in the BDNP's primary architecture has a prior mean, a prior variance, and prior covariances with the other weights corresponding to the same output neuron. To set a fixed proportion $p$ of these to trainable, we find the total number of weights $|\mathbf{W}|$, find the nearest integer to $p|\mathbf{W}|$, and then, proceeding through all weights in the network from the first layer through to the last, we set the prior parameters associated with each weight to trainable until we reach the integer.
> >
> > If the number of datasets was limited and the prior was well-posed, you are correct that it should be beneficial. The difficulty in the data-starved meta-learning regime is that we no longer learn a well-posed prior; we learn a prior that has overfitted to the limited number of training tasks. This problem is exemplified by the BDNP with 1.0 prior learnability on the Abalone setting (2 training tasks) for which we see drastically reduced performance compared to BDNPs with less flexible priors. This is why we advocate for some degree of user-elicitation of priors in this setting; although the resulting prior is unlikely to be perfectly specified, it is much more difficult for it to overfit.
> >
> > > The following works seem relevant to the concept of meta-priors for BNNs and neural processes, and might be useful to include in the related work section: \[...\]
> >
> > Thanks for sharing these papers, we will add them to our bibliography.
> >
> > ### Conclusion
> >
> > We are very grateful for your insightful feedback and questions. We hope that our responses adequately address your concerns, and we would be thrilled if you would consider increasing your score if so. Otherwise, please don't hesitate to reach out with any further questions or concerns you might have. Many thanks once again.
> >
> >
> > [2]. Gordon, J., Bronskill, J., Nowozin, S., Bauer, M., & Turner, R. E. (2019, January). Meta-learning probabilistic inference for prediction. In 7th International Conference on Learning Representations, ICLR 2019.

---

### Official Review · Reviewer_G27C · 2025-11-03

**Soundness:** 3
**Presentation:** 2
**Contribution:** 3
**Rating:** 6
**Confidence:** 3

**Summary:**

This paper introduces a new model called a Bayesian deep neural process (BDNP), which combines aspects of the global inducing point variational posteriors (Ober and Aitchison 2021) and the (latent) neural process (NP) (Garnelo 2018). It is a member of the neural process family, although it differs in that the latent variable is the parameters of the decoder rather than an input to the decoder. Like Ober and Aitchison 2021, the variational posterior uses a psuedo output and a pseudo noise term to exploit the conjugacy of Bayesian linear regression per layer. Unlike Ober and Aitchison 2021, there is a psuedo output and a pseudo noise term for each data point and, instead of treating them as trainiable parameters, the BDNP uses amortization. That is, they are outputs of a trained “inference network”, which is analogous to the encoder of an NP. The model is trained by minimizing a log posterior-predictive density, as in an NP, and a ELBO term, as in VI. In the experiments, the paper shows better KL to the true posterior on a toy example, the ability to meta-learn BNN priors in 1D regression tasks, a comparison of different inference methods under the same meta-learned prior, and an experiment restricting the learned prior.

**Strengths:**

- This is an interesting paper that addresses shortcomings of Bayesian deep learning in a new way, as far as I know. The complexity of the model is both a pro and con.
- I like the idea of amortizing the psuedo observations of Ober and Aitchison 2021
- Figure 6 was interesting. This shows the learned prior is useful regardless of the inference method.
- The experiments focus on meaningful questions, rather than the largest scale experiments

**Weaknesses:**

- There are a few differences with a standard NP, enough that I wonder if this is really an NP (see my questions below).
- Overall, this is a fairly complicated model, which makes it difficult to understand what each component is doing. I appreciated the discussion of the objective function in the appendix, but I think some of this discussion should appear in the main text. The objective doesn’t feel well motivated as it’s written now. More discussion of how this differs from a standard NP would help.
- The introduction discusses how BNNs revert to GPs in the infinite width limit but in the finite width limit considered in this paper, BNNs are not GPs. There are many criticisms to make of BNNs, but I don’t think this one is relevant.
- Unless I missed it, the experiments seem to lack details (architectures, optimizers, etc)
- I found figure 1 (b) a bit confusing. Is this for a one hidden layer network?
- Definitions 1, 2, and 3 aren’t well-defined because they reference a quantity being “small”, but it’s not clear what small means.
- There are a few statements that are too strong for me, e.g. “the impressive performance of our explicitly Bayesian meta-learning setup”, “The BDNP’s approximate posterior is a very good on”

**Questions:**

- Unlike an NP, there is no aggregation of encodings in the BDNP, to ensure permutation invariance. In an NP this is needed to ensure exchangeability of the stochastic process. Instead, the aggregation happens in the loss function. In the appendix, you write that “BDNP and BDAM exhibit this through permutation invariance with respect to the context observations and permutation equivariance with respect to the target inputs”. Can you explain how BDNP defines an exchangeable process without the aggregator?
- This relates to the above question: I’m confused about how exactly the model meta learns the prior. In an NP, the information from the multiple datasets comes from the encoding. The global latent variable is conditioned on the encoding. My understanding is that this is the “meta” learning done in an NP. In the BDNP, the encodings are, I would argue, variational parameters (as in how they are used in Ober and Aitchison 2021). The prior parameters of the BNN are trained similarly to how they would be trained in standard VI, using the fact that the ELBO is a lower bound on the marginal likelihood, which is used for model selection. Note that in an NP, only the log predictive term is in the loss function, not the ELBO term. Can you comment on how the prior meta learning is different from a standard NP? Am I understanding the BDNP correctly?
- You train a separate inference network $g_l$ for each layer, is that correct? This seems like it would create a lot of parameters unless each network is small.
- Can you explain again why the BDNP scales favorably in network width? How do you get around the computation of the mean and covariance in equations 6 and 7 being cubic scaling in the width?

---

> ### Author Response · Authors · 2025-11-28
> **Response to Reviewer G27C's Review. Part 1.**
>
> Many thanks for your thorough review. We are encouraged that you appreciate the scientific merit of our work's motivation and experiments, and that you like our idea to amortise Ober and Aitchison's approximate posterior.
>
> ### Weaknesses
>
> 1. > There are a few differences with a standard NP, enough that I wonder if this is really an NP
>
> This is certainly a valid question to ask. The NP family is both large and diverse, and it is actually quite difficult to define what an NP is. While early NPs (Garnelo-era) consisted of an encoder, aggregator, and decoder, it is not straightforward to disentangle parts of, for example, the ConvCNP or TNP architectures into these three distinct components. Indeed, in the TNP, there is no aggregation at all; each context or target point is represented via a distinct token that is allowed to attend to the other tokens. As discussed in our paper, even permutation equivariance or marginalisation consistency w.r.t. target points are desirable but not absolutely required properties to be an NP, since a number of existing NPs do not satisfy them (TNP-ND, TNP-A, autoregressive CNPs). We would tentatively argue that an NP is any model that maps from a variable-length set of context points and a variable-length of target inputs to a posterior predictive over the corresponding target outputs in an amortised way. The BDNP certainly fits into this description.
>
> If you still feel unsatisfied, we suggest you view the BDNP as Volpp et al. (2021)'s Bayesian context aggregation NP with two modifications: 1.) that a more sophisticated Ober-and-Aitchison-(2020)-style structured variational posterior is used, and 2.) that the decoder takes the context representation as parameters rather than inputs. If we relabel the BNN's weights $\mathbf{W}$ as $\mathbf{z}$, the BDNP's decoder acts on target inputs $\mathbf{X}\_t$ as $d\_\mathbf{z}(\mathbf{X}\_t)$ rather than the usual $d\_\omega(\mathbf{X}\_t, \mathbf{z})$, where $\omega$ denotes some global set of parameters. Note that using just one of these modifications still results in a similar Bayesian-context-aggregation NP; we can amortise Ober and Aitchison's posterior but then pass sampled weights as *inputs* to some more conventional decoder, or we could use samples from a simpler/more traditional amortised approximate posterior to parameterise an MLP which is used as the decoder. Removing either modification brings the model closer to an existing NP family member, but we do not see why either modification should imply departure from the NP family altogether
>
> > Overall, this is a fairly complicated model, which makes it difficult to understand what each component is doing. I appreciated the discussion of the objective function in the appendix, but I think some of this discussion should appear in the main text. The objective doesn’t feel well motivated as it’s written now. More discussion of how this differs from a standard NP would help.
>
> Both you and reviewer `poex` make this request, and we agree that developing the main-text discussion/introduction of the objective function would benefit the paper. We have updated the manuscript to reflect this, and we have uploaded our new manuscript in tandem with these rebuttals.
>
> > The introduction discusses how BNNs revert to GPs in the infinite width limit but in the finite width limit considered in this paper, BNNs are not GPs. There are many criticisms to make of BNNs, but I don’t think this one is relevant.
>
> You are right that a problematic infinite-width limit is not a reason in itself to avoid using standard priors in the finite-width regime. However, since this correspondence indicates that our modelling choices are problematic in an ideal world with infinite compute, it serves as a warning against this choice of prior in general. Indeed, as we use larger finitely-wide nets, the behaviour grows ever closer to that of the corresponding NNGP and the undesirable properties of the infinite-width limit express themselves ever more strongly [1]. So, by pointing to the infinite-width limit, what we are trying to say is something along the lines of: "standard BNN priors are bad, and a theoretical piece of supporting evidence for this is that they turn your BNN into a GP in the infinite-width limit".
>
> > Unless I missed it, the experiments seem to lack details (architectures, optimizers, etc)
>
> This is a totally valid criticism that both you and reviewer `xJWL` make, and we apologise for not including an "Experimental Details" appendix in time for the original submission deadline. We have now completed this appendix and it is visible in our updated manuscript that we uploaded alongside these rebuttals.
>
> > I found figure 1 (b) a bit confusing. Is this for a one hidden layer network?
>
> That is correct, well spotted. We will change the caption to refer to this as a one-hidden-layer BDNP (i.e. to reflect that the fact that two layers of weights corresponds to a single hidden layer of activations).

---

> > ### Author Response · Authors · 2025-11-28
> > **Response to Reviewer G27C's Review. Part 2.**
> >
> > > Definitions 1, 2, and 3 aren’t well-defined because they reference a quantity being “small”, but it’s not clear what small means.
> >
> > It's true that we don't define any threshold beyond which each term in our objective function can be classified as "small". But hopefully we all agree that trained models for which these KL-divergences are smaller are preferable to those for which they are larger. Since proposition 1 just says that our objective function "targets" the desiderata (rather than, e.g., "achieves"), it is not necessary to have a specific definition for "small". If you insist on it, we would be happy to change our definitions to relative ones, for example along the lines of "if KL divergence X is smaller for model A than for model B, then we define model A to be better at Y than model B".
> >
> > > There are a few statements that are too strong for me, e.g. “the impressive performance of our explicitly Bayesian meta-learning setup”, “The BDNP’s approximate posterior is a very good on”
> >
> > We are happy to accommodate this request, and you can see our toned-down versions of these statements in the new manuscript.
> >
> > ### Questions
> >
> > > Unlike an NP, there is no aggregation of encodings in the BDNP, to ensure permutation invariance. In an NP this is needed to ensure exchangeability of the stochastic process. Instead, the aggregation happens in the loss function. In the appendix, you write that “BDNP and BDAM exhibit this through permutation invariance with respect to the context observations and permutation equivariance with respect to the target inputs”. Can you explain how BDNP defines an exchangeable process without the aggregator?
> >
> > The context point encodings (layerwise pseudo-observation and variance pairs) are aggregated in a permutation-invariant manner in the computation of the layerwise posteriors. This aggregation scheme is what makes our approach similar to/an instance of Volpp et al.'s Bayesian context aggregation scheme. Viewing things through the lens of Garnelo-style NPs, equation (4) represents our context data encoding step, and equations (6) and (7) represent the aggregation step of our encoded context points.
> >
> > > This relates to the above question: I’m confused about how exactly the model meta learns the prior. In an NP, the information from the multiple datasets comes from the encoding. The global latent variable is conditioned on the encoding. My understanding is that this is the “meta” learning done in an NP. In the BDNP, the encodings are, I would argue, variational parameters (as in how they are used in Ober and Aitchison 2021). The prior parameters of the BNN are trained similarly to how they would be trained in standard VI, using the fact that the ELBO is a lower bound on the marginal likelihood, which is used for model selection. Note that in an NP, only the log predictive term is in the loss function, not the ELBO term. Can you comment on how the prior meta learning is different from a standard NP? Am I understanding the BDNP correctly?
> >
> > The BDNP's encodings of the context data are what in Ober and Aitchison's work were the variational parameters; you are correct (although in O&A there were $M\ll N$ of these rather than the full set of $N$ that we use). Of course, in the BDNP, they are no longer variational parameters since they are not fixed; they are outputs of inference networks that amortise the task of finding them for each dataset. This means that the variational parameters are now the weights of the inference networks, since these are the *global* parameters that are responsible for defining the variational posterior.
> >
> > To be sure we understand your question correctly, your question boils down to "by what mechanism does the BDNP leverage information in previously observed tasks to make more informed predictions in a new task?"? The answer to this question is: the learned prior. You are correct in saying that, in spirit, our method of learning the prior is very similar to optimising the marginal likelihood. Such an approach can be applied to GPs to learn the best hyperparameters on average over a collection of tasks. The difference between that approach and the BDNP is that we have many more prior parameters (a mean and variance for each weight, as well as covariances between weights corresponding to the same output neuron). In the BDNP, the idea remains to find the prior parameters ("hyperparameters", if you like) that are best on average over a collection of tasks. The prior parameters show up in equations (6) and (7), meaning the information stored in the learned prior (i.e., originally from related tasks) is then used to compute the approximate posterior on a new task.

---

> > > ### Author Response · Authors · 2025-11-28
> > > **Response to Reviewer G27C's Review. Part 3.**
> > >
> > > > You train a separate inference network $g_l$ for each layer, is that correct? This seems like it would create a lot of parameters unless each network is small.
> > >
> > > This is true. As you will see in the new Experimental Details appendix, in most experiments we used the same hidden dimensions in the primary architecture as in the secondary inference networks. It's possible that the earlier layers require shallower inference networks than the later layers, and indeed that the scheme suggested in the footnote on page 2 could allow for much smaller inference networks. However, in our experiments we encountered no problems caused by our inference network architecture choices, so there was no need to experiment with more efficient parameterisations.
> > >
> > > > Can you explain again why the BDNP scales favorably in network width? How do you get around the computation of the mean and covariance in equations 6 and 7 being cubic scaling in the width?
> > >
> > > Apologies for any misunderstanding here. If we can draw your attention to the penultimate paragraph of Appendix C (Minibatching), you will see that the BDNP actually does *not* scale favourably in network width. We also point this out in Section 5 as a reason for why we do not present the BDNP as a practical and general-purpose NP.
> > >
> > > ### Conclusion
> > >
> > > We thank you again for the evident time and effort that went into reviewing our paper, and we are grateful for the positive impact that this will have on our work. We hope that our comments and changes adequately address your concerns, and we remain optimistic that you would maintain or even increase your score as a result. Please let us know if there is anything further you would like to discuss.
> > >
> > > [1]. Aitchison, L. (2020, July). Why bigger is not always better: on finite and infinite neural networks. In Proceedings of the 37th International Conference on Machine Learning (pp. 156-164).

---

### Author Response · Authors · 2025-11-28
**Global Rebuttal Comment**

We would like to express our immense gratitude to all reviewers for the time and effort they have devoted to the peer-review of our work, and we are most pleased with the improvements to our work that they are responsible for. For their benefit and the AC's, we have uploaded a new version of the manuscript simultaneously with our rebuttals---please see the changes we promised to reviewers in red font.

But for belt-and-braces, here we outline the main improvements we have made following the reviewers' recommendations.
1. Comprehensive experimental details are provided in a new appendix, Appendix F.
2. Our objective function is now better motivated during its introduction in section 2.3. We also consistently refer to it as a maximisation objective, rather than a loss. Furthermore, we mention our formal definitions of the three desiderata.
3. The caption to Figure 1b now correctly desribes the BDNP architecture depicted.
4. We have toned-down some of the our language describing the BDNP's performance.
5. We have added pseudocode for the training algorithm of the BDNP. This is found in appendix A.6.
6. We have included references to relevant papers that reviewer Kn36 suggested.
7. We point out in the main text that our within-task minibatching procedure would lead to stochastic optimisation if being used during training. We also change our assertion regarding the type of generalisation that minibatching allows in appendix C to a speculative statement.

We look forward to discussing our work further, and are hopeful that our changes will quell any extant concerns that the reviewers have. Thanks once again.

---

### Meta-Review · Area_Chair_zLji · 2026-01-13

**Summary:**

The paper proposes a neural process-style model that meta-learns a parametric prior over BNN weights per-dataset. Reviewers found the idea interesting and novel, and the theory largely sound. The reviewers raised concerns about clarity and motivation for the training objective, insufficient experimental details, lack of code, and vagueness about "small" in definitions 1-3.

**Reviewer Concerns:**

The rebuttal addresses many concerns, including the motivation for the objective and adds experimental and other details. However, some concerns remain largely unresolved, including the vagueness in definitions 1-3, and the code is still not provided.

**Reviewer Scores:**

Given that many issues were resolved in the rebuttal, it is likely that one or more reviewers would increase their scores, and since the initial scores were already borderline, I recommend accepting the paper.

---

### Decision · Program_Chairs · 2026-01-26

Accept (Poster)